# Towards Better Benchmark Datasets for Inductive Knowledge Graph Completion

## Abstract

Knowledge Graph Completion (KGC) attempts to predict missing facts in a Knowledge Graph (KG). Recently, there's been an increased focus on designing KGC methods that can excel in the *inductive setting*, where a portion or all of the entities and relations seen in inference are unobserved during training. Numerous benchmark datasets have been proposed for inductive KGC, all of which are subsets of existing KGs used for transductive KGC. However, we find that the current procedure for constructing inductive KGC datasets inadvertently creates a shortcut that can be exploited even while disregarding the relational information. Specifically, we observe that the Personalized PageRank (PPR) score can achieve strong or near SOTA performance on most datasets. In this paper, we study the root cause of this problem. Using these insights, we propose an alternative strategy for constructing inductive KGC datasets that helps mitigate the PPR shortcut. We then benchmark multiple popular methods using the newly constructed datasets and analyze their performance. The new benchmark datasets help promote a better understanding of the capabilities and challenges of inductive KGC by removing any shortcuts that obfuscate performance.

## 1 Introduction

Knowledge Graph Completion (KGC) attempts to predict unseen facts given an existing knowledge graph (KG). KGC has numerous applications including drug discovery Zeng et al. (2022), personalized medicine Chandak et al. (2023), and recommendations Wang et al. (2021). Traditionally, most research on KGC was focused on the transductive setting, where the same sets of entities and relations are seen during training and testing. Most methods Bordes et al. (2013); Trouillon et al. (2016); Schlichtkrull et al. (2018) generally focus on learning embeddings for all entities and relations to facilitate the prediction of new facts.

In recent years, interest in KGC has shifted towards designing methods that can generalize to new entities or relations not seen during training. This task, known as "inductive KGC", requires a method to train on a graph $\mathcal{G}_{\text{train}}$ and perform inference on a different graph $\mathcal{G}_{\text{inf}}$, where the inference graph contains either new entities, relations, or both. Because of this, methods for inductive KGC don't rely on fixed embeddings for entities or relations, instead opting for more flexible techniques that can inductively learn representations based on a given graph Teru et al. (2020); Zhu et al. (2021); Lee et al. (2023). To asses the ability of methods for this task, new datasets have been constructed that require methods to reason inductively. All inductive datasets Teru et al. (2020); Galkin et al. (2022); Lee et al. (2023) are constructed from existing transductive KGC datasets. This is done by sampling two graphs, one each for train and inference, which contain disjoint entities. Multiple methods Zhu et al. (2021); Zhang & Yao (2022); Lee et al. (2023) have reported tremendous promise on these newer benchmark datasets.

However, **we find that on almost all inductive datasets, we can achieve competitive performance by using the Personalized PageRank Page et al. (1999) (PPR) score to perform inference**. We note that PPR is a non-learnable heuristic and ignores the relational information in the graph. In Figure 1, we compare the performance of PPR against the supervised SOTA performance on both inductive and transductive datasets. We can see that when performing KGC on inference graphs with either new entities (E) or new entities and relations (E, R), PPR performs only roughly **25%** worse than SOTA. However, this is generally not true for transductive datasets, where PPR usu-

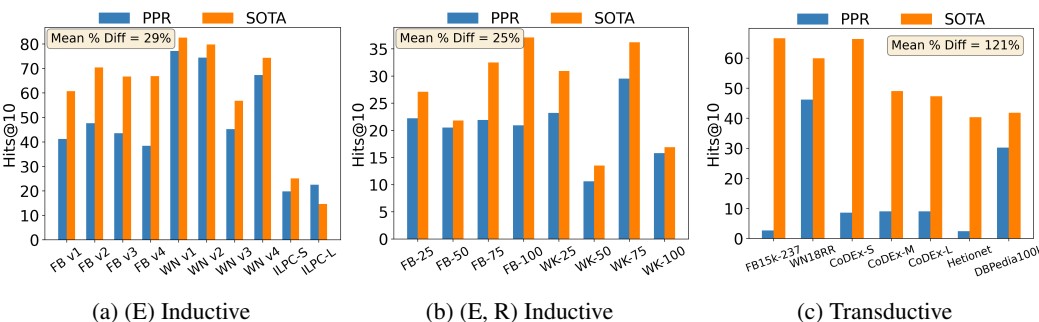

|  (a) (E) Inductive | (b) (E, R) Inductive | (c) Transductive |

Figure 1: Hits@10 of PPR vs. Supervised SOTA. Results are on **(a)** (E) Inductive, **(b)** (E, R) Inductive, and **(c)** Transductive datasets.

ally performs poorly. Interestingly, this observation is true for inductive datasets even when the transductive dataset that it is created from has poor PPR performance. For example, while PPR performs very poorly on FB15k-237, it achieves higher performance on it's inductive derivatives (denoted by "FB"). These findings are problematic as PPR has no basis in literature as a heuristic for KGC, since it completely overlooks the relational aspect of KGs. Therefore, **this suggests the potential existence of a shortcut that allows a simple non-learnable method like PPR to achieve high performance on almost all inductive datasets**. This also brings into question how successful most methods are in inductive reasoning, as a large portion of their performance may be due to this shortcut.

This finding naturally motivates us to ask – *why can PPR perform so well on existing inductive datasets?* In Section 3 we discover that the high performance of PPR is due to how the inductive datasets are created from transductive datasets. Specifically, we observe that the current procedure creates graphs where the shortest path distance (SPD) between entities in positive test samples is much lower than the SPD between those in negative samples. This allows for a method like PPR, which gives a higher weight to shorter walks, to distinguish between the positive and negative samples based solely on the distance. To account for this problem, in Section 4 we propose a new strategy for sampling inductive datasets that uses graph partitioning to create the train and inference graphs. This allows us to sample subgraphs that retain the general properties of the original graph. We then demonstrate that this procedure can indeed create inductive datasets that greatly mitigates the ability of PPR. Lastly, we benchmark common inductive KGC methods on our newly constructed datasets. Our key contributions can be summarized as follows:

- We observe that on existing inductive KGC datasets, we can achieve competitive performance when ranking entities using only the Personalized PageRank score.

- Through empirical study, we find that the strong performance of PPR on inductive datasets is due to the current procedure for constructing inductive datasets, which allows for a shortcut to distinguish between positive and negative samples.

- We propose a new strategy for sampling inductive KGC datasets from their transductive counterparts that uses graph partitioning. We show that our proposed method can substantially mitigate the shortcut. We then benchmark popular methods on our newly created datasets. Compared to the older datasets, methods tend to decrease in performance.

The remaining paper is structured as follows. In Section 2, we provide background on inductive KGC methods and datasets and PPR. In Section 3, we study in detail when and why PPR can perform well on inductive KGC. We then introduce our new strategy for creating inductive datasets in Section 4 and benchmark popular methods on these newly created datasets in Section 5.

## 2 BACKGROUND AND RELATED WORK

Throughout this study we denote a knowledge graph as $\mathcal{G} = \{\mathcal{V}, \mathcal{R}, \mathcal{E}\}$ where $\mathcal{V}$ are the set of entities (i.e., nodes), $\mathcal{R}$ the set of relations (i.e., edge types), and $\mathcal{E}$ the set of edges (i.e., triples) of the form $(s, r, o)$ where $s$ and $o$ are entities and $r$ a relation. Lastly, we note that the task of KGC is formulated as the following where given a query $(s, r, *)$, we attempt to predict the correct entity $*$.

**Inductive KGC Datasets**: In inductive KGC, we are given a training graph $\mathcal{G}_{\text{train}} = \{\mathcal{V}_{\text{train}}, \mathcal{R}_{\text{train}}, \mathcal{E}_{\text{train}}\}$ and an inference graph $\mathcal{G}_{\text{inf}} = \{\mathcal{V}_{\text{inf}}, \mathcal{R}_{\text{inf}}, \mathcal{E}_{\text{inf}}\}$. A method is trained on $\mathcal{G}_{\text{train}}$ and evaluated on $\mathcal{G}_{\text{inf}}$. Most datasets consider the setting that is only disjoint on the entities such that $\mathcal{R}_{\text{inf}} \subseteq \mathcal{R}_{\text{train}}$ and $\mathcal{E}_{\text{inf}} \cap \mathcal{E}_{\text{train}} = \emptyset$. This setting is referred to as the **(E) setting**. Another setting, which we denote as **(E, R)** further allows $\mathcal{R}_{\text{inf}}$ to contain relations not in $\mathcal{R}_{\text{train}}$.

All existing inductive datasets are sampled from existing transductive datasets. Furthermore, the majority of inductive datasets Teru et al. (2020); Lee et al. (2023) are further created via the same procedure introduced by Teru et al. (2020). We now give a brief overview of this procedure. Given a transductive dataset, which we denote as $\mathcal{G}$, $k$ seed entities are randomly chosen from $\mathcal{G}$. The 2-hop neighborhood is then extracted around each individual seed entity. To prevent exponential growth, the number of neighbors sampled at any hop is capped at 50 for each seed entity. The resulting edges are then combined to create $\mathcal{G}_{\text{train}}$ and are subsequently removed from the original graph. The inference graph is then sampled in a similar manner from the resulting graph $\mathcal{G} \backslash \mathcal{G}_{\text{train}}$. One exception to this procedure are the ILPC datasets, introduced by Galkin et al. (2022), which instead sample $p\%$ of the nodes from $\mathcal{G}$ and use them to create $\mathcal{G}_{\text{train}}$. The rest of the nodes are then used to construct $\mathcal{G}_{\text{inf}}$. In practice, we find that both methods tend to produce similar graphs.

**Inductive KGC Methods**: NeurlLP Yang et al. (2017) and DRUM Sadeghian et al. (2019) consider combining the path representations of different length between both entities in a triple. However, since they explicitly consider each path, they are often limited to only considering paths of up to length 2 or 3. Conditional MPNNs Huang et al. (2024) are a more efficient alternative to encoding higher-order path information. They work by conditioning the message passing mechanism on the known entity, allowing the implicit aggregation of all paths up to a length $L$ (which is equal to the number of GNN layers). The value of $L$ is typically $5/6$. Prominent examples include NBFNet Zhu et al. (2021) and RED-GNN Zhang & Yao (2022). More scalable alternatives have been proposed that prune the propagated messages Zhu et al. (2022); Zhang et al. (2023); Shomer et al. (2023b). NodePiece Galkin et al. (2021) is concerned with parameter efficiency, using an anchor-based approach to learn a more compact set of entity and relation representations. All previous methods assume that a fixed set of relations exist between the train and inference graphs. To account for new relations in inference, InGRAM Lee et al. (2023) introduces the concept of a "relation graph", which inductively encodes the representation of each relation. Gao et al. (2023) introduces the concept of "double permutation- equivariant representations" as a way to model KGs that are equivariant to permutations of both the entities and relations. They theoretically show that capturing this property is essential for proper generalization across KGs. To this point these introduce a new methodIS-DEA/ISDEA+ that can satisfy this property. They further introduce a variant of InGRAM Lee et al. (2023), DEq-InGram, that endows it with the ability to compute double equivariant representations. Note that we omit subgraph methods that have been to shown to prohibitively expensive Teru et al. (2020); Liu et al. (2021); Mai et al. (2021); Xu et al. (2022); Geng et al. (2023) or those that require the use of external textual information Gesese et al. (2022); Daza et al. (2021).

**Personalized PageRank**: PageRank Page et al. (1999) computes the probability of finishing a random walk of arbitrary length at some node $u$, when there is equal probability of beginning the walk at any node in the graph. Personalized PageRank (PPR) Page et al. (1999) is a version of PageRank that is "personalized" to some root node $s$, where at each step there is a probability $\alpha$ of teleporting to $s$. The set of PPR scores for a root node $s$ is given by $\text{pr}_s \in \mathbb{R}^{|\mathcal{V}|}$ and can be formulated as the weighted sum of all random walk probabilities between two nodes Chung (2007):

$$\text{pr}_s = \alpha \sum_{k=0}^{\infty} (1 - \alpha)^k W^k x_s, \tag{1}$$

where $W = D^{-1}A$ and $x_s$ is a one-hot vector at node $s$. Observe that the weight given to a walk decays with the increase in length due to $(1 - \alpha)^k$. As such, the PPR score will often be higher for those nodes of shorter distance to $s$. For a KG, we obtain the PPR matrix by first converting the inference graph, $\mathcal{G}_{\text{inf}}$, to an undirected graph. This is common practice in KGC Dettmers et al. (2018) whereby inverse relations are added to the graph. We further assume an edge weight of 1 for all edges. As such, *the relations are completely disregarded when computing the PPR*. Please see Appendix C for a more detailed discussion of PPR and how it is used in our paper.

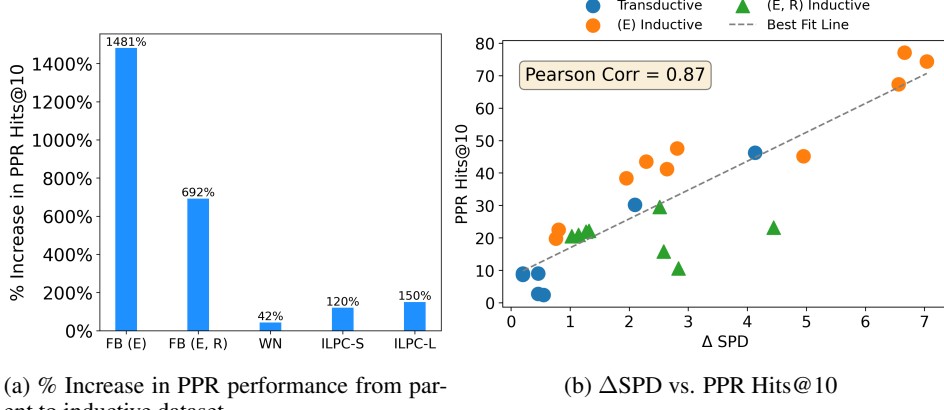

(a) % Increase in PPR performance from parent to inductive dataset

(b) $\Delta$SPD vs. PPR Hits@10

Figure 2: (a) Percent increase in PPR performance on the inductive datasets as compared to their parent transductive dataset. For datasets with multiple splits (i.e., FB and WN) we take the mean performance across each. (b) Relationship between $\Delta$SPD and performance. We observe a strong relationship among both transductive and inductive datasets.

## 3 PRELIMINARY STUDY

In this section, we examine the performance on common inductive KGC datasets. We first show that the Personalized PageRank (PPR) can often serve as a good baseline on most datasets. From this observation, we attempt to answer two important questions: (1) When can the PPR scores perform well on KGC? and (2) Why can PPR sometimes perform well for KGC?

### 3.1 PERSONALIZED PAGERANK (PPR) IS A STRONG BASELINE FOR INDUCTIVE KGC

We begin by obtaining the PPR matrix of the inference graph for use in evaluation. See Section 2 for more details on how this is done. Given this new graph, for a query $(s, r, *)$, we calculate the PPR score for all possible entities $o \in \mathcal{V}$. Using these scores, we can obtain the rank of the true entity for our query. We emphasize that PPR is (a) a non-learnable heuristic, (b) ignores the relations in the KG, and (c) has no basis in KG literature as a method to perform KGC.

In Figure 1, we show the performance when using the PPR versus the SOTA performance among supervised methods. The SOTA method is dataset dependent, with the specific methods listed in Appendix A.1. We split the datasets by transductive, (E), and (E, R) inductive. We include those datasets most often used in each task, comprising 25 in total. Please see Appendix B.1 for more on the datasets chosen. The full set of results can also be found in Appendix A.1.

We observe that on both types of inductive datasets, the PPR score does reasonably well, **performing on average only 25-29% less than SOTA**. This is surprising as PPR is both non-learnable and ignores the relational aspect of the KG. We also find that on some datasets like the WN or ILPC inductive splits, the performance nearly matches or exceeds the supervised performance. On the other hand, for the transductive datasets, the performance disparity is often much larger. Interestingly, we note that PPR still performs well on some transductive datasets, including the popular WN18RR. This tells us that this phenomenon is not necessarily unique to inductive datasets, but is most apparent there.

Furthermore, we find that for the inductive datasets, their PPR performance is much higher than their transductive parents. We detail this in Figure 2a where for different inductive datasets we see the percent increase in PPR performance from transductive to inductive. For example, on FB15k-237 the PPR Hits@10 is 2.7%, however the mean performance on the four FB (E) splits is 42.7%, representing a 1481% increase. We can see that the % increase on each inductive dataset is large, with the smallest being 42% on WN. This suggests that there is some change in the underlying inductive graphs that are causing the performance of PPR to increase. Lastly, we further explore other potential shortcuts in Appendix A.3, finding that PPR is by far the most severe.

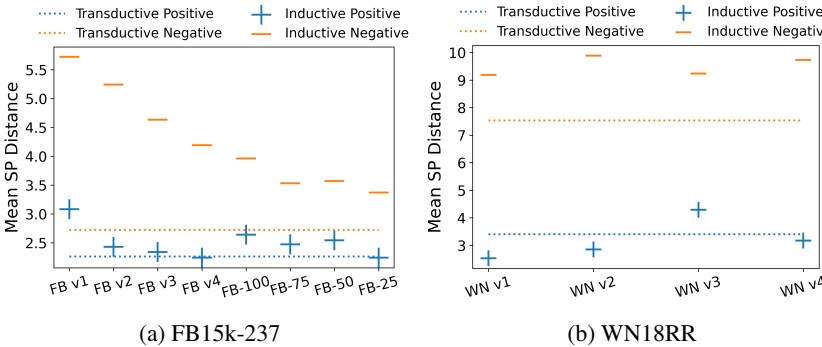

(a) FB15k-237             (b) WN18RR

Figure 3: The mean shortest path (SP) distance between positive test samples and negative samples. We display the results for two transductive datasets (FB15k-237 and WN18RR) and the inductive datasets that are derived from them.

## 3.2 WHEN DOES PPR PERFORM WELL AND WHY?

In the previous section we detailed that PPR score performs well on the inductive setting. Furthermore, the performance of PPR on inductive datasets is much higher than their transductive counterparts of which they're derived from. This raises the question – *why can PPR perform well on some datasets but not others?*

We find that the answer lies in **comparing the mean shortest path distance (SPD) between positive and negative samples**. For a query $(s, r, *)$, let's denote $o^+$ as the correct answer and $o^- \in \mathcal{V}^-(s, r)$ as the set of negative answers for the query. We compute the SPD on the inference graph for both the $o^+$ and $\mathcal{V}^-(s, r)$. This is repeated for each test sample $(s, q, *) \in \mathcal{E}_{\text{inf}}$. We then calculate the mean SPD across all positive and negative test samples, which we denote as $\text{SPD}^+$ and $\text{SPD}^-$, respectively. The difference in mean SPD is correspondingly given by $\Delta\text{SPD} = \text{SPD}^- - \text{SPD}^+$. This tells us, on average, how much shorter the distance between entities in positive samples are relative to those in negatives.

In Figure 2b we show the relationship between $\Delta\text{SPD}$ and the Hits@10 when using PPR. The results are across 25 different datasets, for both inductive and transductive KGC. Calculating the Pearson correlation, we can observe it is 0.87, which indicates a strong relationship between the two metrics. Therefore, there exists a basic pattern to distinguish positive and negative samples in many KGC datasets. Put simply, the SPD between entities in positive test samples tend to be lower than that in negative samples. The larger the discrepancy between the mean distances, the better PPR can perform. This suggests that the PPR scores can exploit this pattern in the datasets, to achieve good performance, even while completely ignoring the relational information.

*But, why can PPR exploit this pattern?* As shown in Eq. equation 1, the PPR score between two nodes is the weighted sum of walks between them. Furthermore, walks of shorter length are weighed more heavily than those of longer length. Therefore, nodes with a shorter distance between them will be able to benefit from these highly-weighted walks, while those of larger distance will not. For example, when $\alpha = 0.15$, the highest weight for a walk when SPD $= 2$ is $0.72$ while when SPD $= 5$ it is $0.44$. As such, the PPR score will invariably favor those node pairs with a lower SPD.

Note that we use PPR as opposed to SPD in our experiments since all-pairs SPD is costly to calculate and the full PPR matrix can be efficiently approximated via Andersen et al. (2006). Furthermore, as we show in Section 3.3, even when controlling for the SPD, the PPR score can help differentiate between positive and negative samples.

## 3.3 WHY IS THIS SO COMMON ON INDUCTIVE DATASETS?

In the last section, we covered when PPR can perform well on KGC and why. However, one remaining question is – *why is this trend so pervasive on inductive datasets but rarely on their transductive counterparts?* For example, on FB15k-237, a transductive dataset, the Hits@10 of PPR is 2.7%.

However, across eight different inductive versions of FB15k-237, the mean PPR performance is 32%. We find that this can be explained by the following two observations.

**Observation 1: The current procedure for creating inductive datasets increases the $\Delta$SPD and thereby the performance of PPR.** As noted earlier, all common inductive datasets are created from existing transductive datasets. See Section 2 for a detailed overview of the construction process. In a nutshell, the training and inference graph are constructed sequentially by sampling a number of subgraphs from a graph. In Figure 3 we show the mean SPD of both positive and negative samples for 12 inductive datasets and their parent transductive dataset. We limit our analysis to those datasets derived from FB15k-237 and WN18RR, as the majority of inductive datasets are derived from them. We observe that for all inductive datasets, while the mean SPD for negative samples sharply rises, the mean SPD for positive samples stay roughly the same. This creates the shortcut described in Section 3.2, where the SPD can easily differentiate between positive and negative samples. Since the gap between the distances is almost always larger than the original transductive dataset, this shortcut becomes more pronounced, thereby leading to a better PPR performance (see Appendix A.1 for detailed results).

But why does the mean SPD drastically change for negative samples but not positive? We find that entities in positive samples are more well-connected those those in negatives. In Table 1, we show the % difference in PPR score between positive and negative samples when controlling for the SPD on two inductive datasets.

We can see that the PPR score is much greater for positive samples, even for higher values of SPD. Therefore, the SPD of positive samples are better able to "withstand" changes in the underlying graph better than negatives, as they typically contain additional shorter walks between samples. However, negatives are typically much less well-connected, so they are more affected by removing a portion of edges from the original graph.

Table 1: % Increase in Positive vs. Negative sample PPR, broken down by SPD.

| SPD | WN18RR v4 | FB15k-237 v4 |
|---|---|---|
| $[1, 2)$ | +17% | +5% |
| $[2, 3)$ | +29% | +200% |
| $[3, 4)$ | +82% | +328% |
| $[4, \infty)$ | +2837% | +44% |

**Observation 2: Constructing a good inductive dataset is difficult.** In Section 2 we discuss the general algorithm used to create inductive datasets. Multiple parameters exist that guide the construction process. It is tempting to think that by simply trying different combinations, one can happen upon a split that doesn't suffer from high PPR performance. However, in practice, we find that this is difficult, as there are multiple factors to contend with.

We demonstrate this by attempting to generate inductive datasets from FB15k-237. We generate a number of different inductive datasets by modifying the **(a)** # of seed entities for train and inference and **(b)** the maximum neighborhood size for train and inference. For different combination of values, we generate three different datasets using different random seeds. For each of the generated datasets, we calculate the size of both the train and inference graphs, the $\Delta$SPD, and the PPR performance. These values are then averaged across seeds. A more detailed discussion is given in Appendix D. In Figure 4a we plot the $\Delta$SPD vs. the PPR performance. Despite searching across a wide variety of parameters, both $\Delta$SPD and the PPR performance remain much higher for the inductive datasets compared to the transductive dataset. Furthermore, we show the relationship between the size of the train and inference graphs and the PPR performance in Figures 4b and 4c. We observe that when the performance of PPR is at its lowest, the size of the train graph is noticeably small. However, fixing this problem, results in a sharp increase in the performance of PPR. Furthermore, there is an inverse relationship between the size of the train and inference graph, making it hard to find a "sweet spot" where the PPR performance is low and both graphs aren't too small.

The studies in this section show that the current strategy for constructing inductive datasets from existing transductive datasets is liable to introduce a well-performing shortcut into the existing graph. Currently, it almost always leads to a sharp increase in the PPR performance. Furthermore, attempting to limit the severity of this issue is very difficult while also generating train and inference graphs of reasonable sizes. **This suggests that a new strategy is needed for sampling graphs for inductive KGC from transductive datasets.**

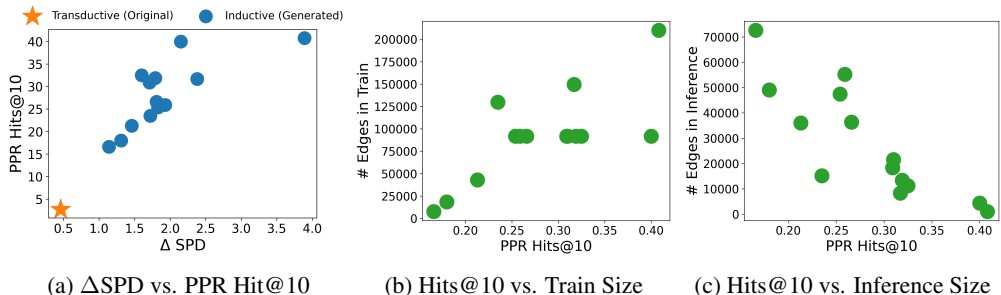

(a) ΔSPD vs. PPR Hit@10     (b) Hits@10 vs. Train Size     (c) Hits@10 vs. Inference Size

Figure 4: Experiments when generating inductive datasets from FB15k-237. (a) The generated datasets always have a much higher ΔSPD than the transductive dataset. (b)/(c) There is a trade-off between the size of the train and inference graph, making it difficult to obtain a good split.

## 4   CONSTRUCTING INDUCTIVE DATASETS THROUGH GRAPH PARTITIONING

In the previous section, we showed that the PPR score can achieve strong performance on most inductive KGC datasets. Furthermore, we demonstrate that this is due to how inductive datasets are sampled from existing transductive datasets. This sampling strategy engenders a shift in the underlying properties of the graph that allows for PPR to perform well. This naturally causes us to ask – *How can we mitigate this problem when constructing newer inductive datasets?* In the next subsection, we introduce our strategy which utilizes graph partitioning to alleviate this problem.

### 4.1   PARTITION-BASED DATASET SAMPLING

We've previously covered in Section 3 that the existing procedure for creating inductive datasets leads to suboptimal subgraphs. This is because the resulting subgraphs tend to have much different properties than the original graph, such as the distance distribution, which can potentially lead to shortcuts when performing KGC.

We note that the task of constructing inductive datasets from an existing graph $\mathcal{G}$ can be framed as a graph partitioning problem. Formally, we want to sample two non-overlapping partitions from the graph such that $\mathcal{G}_{\text{train}}, \mathcal{G}_{\text{inf}} \subseteq \mathcal{G}$ and $\mathcal{G}_{\text{train}} \cap \mathcal{G}_{\text{inf}} = \emptyset$. Given the analysis in Section 3.2, we hope to sample subgraphs such that $\Delta\text{SPD}(\mathcal{G}_{\text{train}}) \approx \Delta\text{SPD}(\mathcal{G}_{\text{inf}}) \approx \Delta\text{SPD}(\mathcal{G})$. But, *how do we find subgraphs that satisfy this property?* Intuitively, we want to sample each subgraph so that it's removal has little effect on the initial graph's structure. We give an example in Figure 5 where we sample two subgraphs from an existing graph. As we can see, even though the graph is partitioned, the relationship between entities *in the same partition* remain roughly the same before and after the partition. This is because there already exists little relationship between the two partitions in the original graph.

Multiple popular approaches Shi & Malik (2000); Blondel et al. (2008) exist that attempt to divide the graph into optimal partitions. The guiding principle in these approaches is that the partitions should be internally dense but only sparsely connected to one another. Because of this, the entities in different partitions should only be weakly connected and have little impact on one another. Therefore, the relationship between entities in the same partition are minimally affected by outside entities or

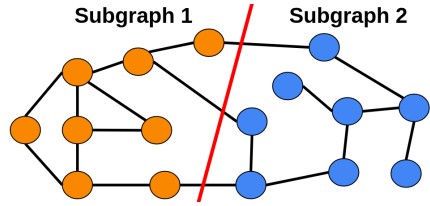

Figure 5: Sampling Two Subgraphs

edges. As such, removing this partition from the graph should then have little effect on the entities in that partition. Also, since the partitions are created at the same time, we avoid sampling the inference graph after the training, which can be suboptimal.

In practice, we consider using Spectral Clustering Shi & Malik (2000) or the Louvain method Blondel et al. (2008), as dependent on the dataset. Once the original graph is partitioned into $n$ partitions, we sample $k$ of those to be used. The partitions are chosen such that they display similar properties to the original graph (see Section 4.2 for more). Of the $k$ partitions, one is chosen as the training

Table 2: Mean ΔSPD and PPR Hits@10 of new and old Inductive ("Ind.") datasets vs. their Transductive ("Trans.") parent. We highlight the value of the inductive dataset closer to the transductive as **blue**. For every case, the ΔSPD and PPR Hits@10 of the new splits are more aligned with the original transductive dataset.

| Task | Dataset | Mean PPR Hits@10 | | | Mean ΔSPD | | |
|---|---|---|---|---|---|---|---|
| | | Trans. | Ind. Old | Ind. New | Trans. | Ind. Old | Ind. New |
| (E) | WN18RR | 46.2 | 66.0 | **45.1** | 4.1 | 6.3 | **3.2** |
| | CoDEx-M | 9.0 | 21.1 | **11.2** | 0.2 | 0.78 | **0.24** |
| | HetioNet | 2.4 | **NA** | 2.7 | 0.55 | **NA** | 0.29 |
| (E, R) | FB15k-237 | 2.7 | 21.4 | **10.8** | 0.46 | 2.42 | **0.48** |
| | CoDEx-M | 9.0 | **NA** | 13.2 | 0.20 | **NA** | 0.42 |

graph while the other $k-1$ are designated as *separate* inference graphs. This is an important advantage of using partitioning to sample the graphs, as it allows us to avoid having to sample multiple different train and inference pairs as in Teru et al. (2020); Lee et al. (2023), thereby allowing for more efficient benchmarking. Lastly, we note that when sampling graphs for the (E) inductive task, we further remove new relations from the inference graph. In practice, we find that we can easily find partitions where this amounts to little or no change in the inference graph. See Section B.2 for more details on the dataset creation process.

## 4.2 ANALYSIS OF NEW DATASETS

In this section we analyze the new inductive datasets created following the partition-based procedure outlined in Section 4.1. We create (E) inductive datasets, from WN18RR Dettmers et al. (2018), CoDEx-M Safavi & Koutra (2020), and HetioNet Himmelstein et al. (2017). For the (E, R) datasets, we use CoDEx-M Safavi & Koutra (2020) and FB15k-237 Toutanova & Chen (2015). Note that some datasets are only suitable for one task or another. For example, WN18RR and HetioNet have very few relations, making it nearly impossible to sample two partitions with significantly different relations for the (E, R) task. On the other hand, FB15k-237 contains too many relations to sample multiple graphs for the (E) tasks without removing many edges from either graph.

In Table 2 we show the PPR Hits@10 and ΔSPD for the inductive datasets and their original transductive dataset. When multiple inference graphs exist, we take the mean across each inference graph. When possible, we also include a comparison against those inductive datasets that already exist. For example, for WN18RR in the (E) task, 4 datasets exist from Teru et al. (2020). Compared to the old inductive datasets, the PPR performance for the newer datasets is much lower. Specifically, the **average PPR performance is 78% lower on the new inductive datasets as compared to the older datasets**. Also, the PPR performance of the new inductive datasets is very similar to the performance on the original transductive dataset. A similar trend can be found when comparing the ΔSPD. This analysis shows that newer sampling procedure can indeed sample inductive datasets that are much more similar to the original transductive graph, **greatly mitigating the PPR shortcut**.

## 5 EXPERIMENTS

### 5.1 EXPERIMENTAL SETTINGS

**Datasets**: We use the new datasets created in Section 4.1. For the (E) setting, this includes WN18RR, HetioNet, and CoDEx-M. For the (E, R) setting, it is FB15k-237 and CoDEx-M. We sample 2 inference graphs for each dataset, except for CoDEx-M on the (E) setting where we could only find 1 suitable graph for inference. For each inference graph, 10% of edges are randomly removed for testing. For validation 10% of edges are removed from the training graph. It is necessary that the validation samples are extracted from the train graph as *the inference graphs must remain unobserved during training*. The statistics for each dataset can be found in Appendix B.2.

**Baseline Methods**: We consider prominent KGC methods including NBFNet Zhu et al. (2021), RED-GNN Zhang & Yao (2022), NodePiece Galkin et al. (2021), InGram Lee et al. (2023), DEq-InGram Gao et al. (2023), and Neural LP Yang et al. (2017). We also consider the recent foundation

Table 3: (E) Inductive Results (Hits@10) for supervised methods.

| Models | CoDEx-M | WN18RR | | HetioNet | |
| --- | --- | --- | --- | --- | --- |
| | Inference 1 | Inference 1 | Inference 2 | Inference 1 | Inference 2 |
| PPR | 11.2 | 66.2 | 24 | 3.2 | 2.2 |
| Neural LP | 13.0 ± 17.9 | 37.9 ± 1.4 | 14.8 ± 1.9 | 12.0 ± 16.4 | 10.7 ± 15.0 |
| NodePiece | 6.8 ± 0.8 | 29.6 ± 0.8 | 4.8 ± 0.6 | 10.2 ± 0.9 | 15.4 ± 0.9 |
| InGram | 20.1 ± 3.5 | 38.0 ± 2.4 | 8.0 ± 2.9 | 21.9 ± 1.1 | 22.3 ± 2.8 |
| DEq-InGram | 23.8 ± 1.6 | 62.5 ± 0.8 | 19.1 ± 3.1 | 26.5 ± 4.1 | 28.8 ± 3.5 |
| RED-GNN | 35.6 ± 2.3 | 72.9 ± 0.4 | 27.7 ± 0.3 | 68.3 ± 3.0 | **85.1 ± 2.7** |
| NBFNet | **43.6 ± 0.2** | **75.5 ± 0.2** | **29.4 ± 2.5** | **72.8 ± 3.8** | 77.2 ± 0.4 |

Table 4: (E, R) Inductive Results for supervised methods. The % of new relations are in parentheses.

| Models | FB15k-237 | | CoDEx-M | |
| --- | --- | --- | --- | --- |
| | Inference 1 (27%) | Inference 2 (63%) | Inference 1 (10%) | Inference 2 (57%) |
| PPR | 9.1 | 12.4 | 10.9 | 15.4 |
| Neural LP | 17.5 ± 9.9 | 22.4 ± 12.8 | 16.7 ± 22.8 | 9.8 ± 13.7 |
| NodePiece | 3.0 ± 0.6 | 4.7 ± 0.5 | 3.1 ± 0.6 | 2.5 ± 1.0 |
| InGram | 23.8 ± 3.0 | 20.2 ± 2.0 | 20.4 ± 3.3 | 15.9 ± 10.0 |
| DEq-InGram | **35.4 ± 2.5** | 27.1 ± 3.5 | 35.2 ± 14.4 | 24.7 ± 0.9 |
| RED-GNN | 21.6 ± 5.8 | **33.3 ± 4.2** | 29.2 ± 2.9 | **26.5 ± 10.4** |
| NBFNet | 27.5 ± 1.8 | 26.2 ± 0.3 | **47.7 ± 11.8** | 17.6 ± 10.0 |

model ULTRA Galkin et al. (2023). We omit methods that have been shown to either be prohibitively expensive (e.g., Teru et al. (2020); Liu et al. (2021); Mai et al. (2021)) or require the use of textual information Gesese et al. (2022); Daza et al. (2021).

The full set of experimental settings can be found in Appendix F.

## 5.2 RESULTS

**Main Results**: The main results for the supervised methods can be found in Tables 3 and 4. We observe that on nearly every dataset, NBFNet and RED-GNN are the two best models. This indicates that conditional MPNNs Huang et al. (2024), the class of model in which both belong to, are necessary for strong performance in inductive KGC. Interestingly, we observe that InGram struggles in the (E, R) setting. This is even true when the % of new relations is high. This runs counter to the results on older inductive datasets Lee et al. (2023) where InGram excelled over NBFNet and RED-GNN. However, this is not true for DEq-InGram, which performs consistently well under the (E, R) setting. Lastly, we observe that Neural LP sometimes fails to converge, resulting in a near zero performance and thus causing the model to have a high performance variance.

**Performance Comparison of PPR vs. SOTA**: In our original analysis, we showed that for the older inductive datasets, the performance gap between the SOTA method of each dataset and PPR is quite small. Specifically, the mean % difference between the two was relatively small at 25-29% (see Figure 1). We now perform the same analysis on the new inductive datasets. The results are shown in Figure 6a. We observe that PPR generally performs much worse than the SOTA method with a mean % difference of 101%. Furthermore, this is more in line with the transductive datasets which have a mean % difference of 121%. This suggests that in addition to the decrease in raw PPR performance, the relative performance of PPR also decreases on the newer datasets.

**Performance Comparison on New and Old Inductive Datasets**: In Table 2, we compared both the PPR performance and ΔSPD on the new and older inductive datasets. We found that both metrics tend to be much higher on the older inductive datasets, indicating that our newer splits are effective in mitigating the shortcut. Given those results, a natural question is *whether we see a similar drop in performance for neural methods?* We limit our analysis to WN18RR (E) and FB15k-237 (E, R). This is either due to a lack of older datasets (i.e., CoDEX-M (E, R) and HetioNet (E)) or minimal results on the older datasets, i.e., CoDEX-M (E). For each dataset, we compute the mean performance across inference graphs. The results are shown in Table 5. We find that performance

Table 5: Mean performance on Old vs. New inductive datasets by Method. **Red** indicates a decrease in the mean performance on the newer splits.

| Models | WN18RR (E) | | | FB15k-237 (E, R) | | |
|---|---|---|---|---|---|---|
| | Old Ind. | New Ind. | % Difference | Old Ind. | New Ind. | % Difference |
| PPR | 66.0 | 45.1 | **-31.7%** | 21.4 | 10.8 | **-49.5%** |
| Neural LP | 67.6 | 26.4 | **-61.0%** | 16.2 | 20.0 | +23.1% |
| NodePiece | 29.8 | 17.2 | **-42.3%** | 5.0 | 3.9 | **-23.0%** |
| InGram | 49.9 | 23.0 | **-53.9%** | 29.6 | 22.0 | **-25.7%** |
| RED-GNN | 70.6 | 50.3 | **-28.8%** | 25.0 | 27.5 | +9.8% |
| NBFNet | 72.2 | 52.5 | **-27.4%** | 24.8 | 26.9 | +8.3% |
| **Mean** | - | - | **-40.6%** | - | - | **-9.5%** |

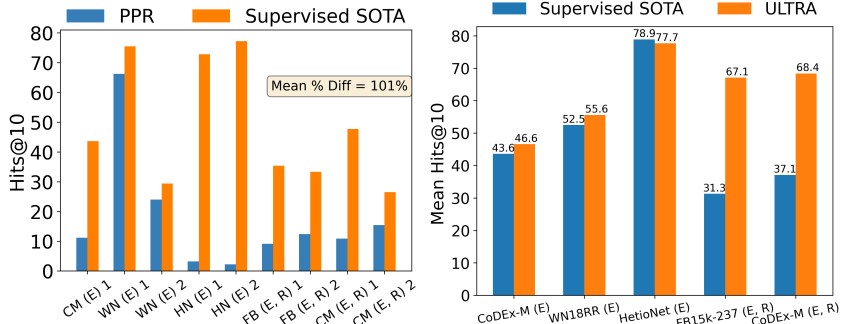

(a) PPR vs. SOTA Hits@10 on new datasets  (b) Mean Hits@10 for ULTRA vs. SOTA

Figure 6: (a) Performance of PPR vs. SOTA on new inductive datasets. Note that we abbreviate CoDEx-M as CM, FB15k-237 as FB, HetioNet as HN and WN18RR as WN. Furthermore, the number represents the inference graph. (b) Mean performance of the foundation model ULTRA Galkin et al. (2023) vs. the supervised SOTA on the newer datasets.

of most methods drops significantly on both datasets, with an average drop of 40.6% and 9.5% on WN18RR (E) and FB15k-237 (E, R), respectively. **This suggests that removing the shortcut has a large negative effect on the performance**, suggesting that our new datasets are indeed harder.

**Performance of ULTRA** Galkin et al. (2023): We further compare against ULTRA, a recent foundation model designed for fully inductive KGC. We evaluate ULTRA under the 0-shot setting. Since the setting of ULTRA differs from that of the other methods (i.e., 0-shot), we display it separately from the other methods in Tables 3 and 4. See Appendix H for more details on the versions of ULTRA used. The results are in Figure 6b where for each dataset, we average the results across the different inference graphs (full results in Appendix A.2). On the (E) task, ULTRA is comparable to the supervised SOTA. However, on the (E, R) task, ULTRA significantly outperforms other methods. This suggests that ULTRA contains a greater generalization ability than other methods.

## 6 CONCLUSION

In this paper we study the problem of constructing datasets for inductive knowledge graph completion. Upon examination, we find that we can achieve competitive performance on most inductive datasets through the use of Personalized PageRank Page et al. (1999), which ignores the relational structure of the graph. Through our study, we uncover that this shortcut is due to how inductive datasets are created. To remedy this problem, we propose a new dataset construction process based on graph partitioning that empirically mitigates the impact of the studied shortcut. We then construct new benchmark datasets using this new procedure and benchmark various methods. Examining the results, we observe that the relative performance decreases on most datasets and methods, indicating that the newer benchmark datasets are harder than the previous ones. For future work, we plan to explore creating inductive KG datasets that aren't sampled from existing transductive datasets.

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

## A  ADDITIONAL RESULTS

### A.1  SUPERVISED SOTA VS. PPR PERFORMANCE

In this section we show the performance in terms of Hits@10 for the SOTA supervised method vs. the Personalized PageRank (PPR) score. We further include the % difference in performance and which method is considered SOTA. These are shown in Tables 6, 7, and 8 for the (E), (E, R), and transductive datasets, respectively. For an overview of SOTA performance on additional KG datasets, please see Galkin et al. (2023).

Table 6: Hits@10 for PPR vs. Supervised SOTA on the **(E) Inductive Datasets**

| Dataset | Supervised SOTA | PPR | % Difference | SOTA Method |
|---------|-----------------|------|--------------|-------------|
| WN v1 | 82.6 | 77.1 | 7% | NBFNet Zhu et al. (2021) |
| WN v2 | 79.8 | 74.4 | 12% | NBFNet Zhu et al. (2021) |
| WN v3 | 56.8 | 45.2 | 26% | NBFNet Zhu et al. (2021) |
| WN v4 | 74.3 | 67.3 | 10% | A*Net Zhu et al. (2022) |
| FB v1 | 60.7 | 41.2 | 47% | NBFNet Zhu et al. (2021) |
| FB v2 | 70.4 | 47.6 | 48% | NBFNet Shomer et al. (2023b) |
| FB v3 | 66.7 | 43.5 | 53% | NBFNet Zhu et al. (2021) |
| FB v4 | 66.8 | 38.4 | 74% | NBFNet Zhu et al. (2021) |
| ILPC-S | 25.1 | 19.8 | 27% | NodePiece Galkin et al. (2021) |
| ILPC-L | 14.6 | 22.5 | -35% | NodePiece Galkin et al. (2021) |

Table 7: Hits@10 for PPR vs. Supervised SOTA on the **(E, R) Inductive Datasets**

| Dataset | Supervised SOTA | PPR | % Difference | SOTA Method |
|---------|-----------------|------|--------------|-------------|
| FB-100 | 37.1 | 22.2 | 67% | InGram Lee et al. (2023) |
| FB-75 | 32.5 | 21.9 | 48% | InGram Lee et al. (2023) |
| FB-50 | 21.8 | 20.5 | 6% | InGram Lee et al. (2023) |
| FB-25 | 27.1 | 20.9 | 30% | InGram Lee et al. (2023) |
| WK-100 | 16.9 | 15.8 | 7% | InGram Lee et al. (2023) |
| WK-75 | 36.2 | 29.5 | 23% | InGram Lee et al. (2023) |
| WK-50 | 13.5 | 10.6 | 27% | InGram Lee et al. (2023) |
| WK-25 | 30.9 | 23.2 | 33% | InGram Lee et al. (2023) |

Table 8: Hits@10 for PPR vs. Supervised SOTA on the **Transductive Datasets**

| Dataset | Supervised SOTA | PPR | % Difference | SOTA Method |
|---------|-----------------|------|--------------|-------------|
| FB15k-237 | 66.6 | 2.7 | 2367% | NBF+TAGNet Shomer et al. (2023b) |
| WN18RR | 59.9 | 46.2 | 30% | NBF+TAGNet Shomer et al. (2023b) |
| CoDEx-M | 49.0 | 9.0 | 444% | ComplEx RP Chen et al. (2021) |
| CoDEx-S | 66.3 | 8.6 | 671% | ComplEx RP Chen et al. (2021) |
| CoDEx-L | 47.3 | 9.0 | 426% | ComplEx RP Chen et al. (2021) |
| Hetionet | 40.3 | 2.4 | 1579% | RotatE Sun et al. (2019) |
| DBPedia100k | 41.8 | 30.2 | 38% | ComplEx-NNE+AER Ding et al. (2018) |

### A.2  PERFORMANCE OF ULTRA

We include the full results of ULTRA Galkin et al. (2023) on each dataset and inference graph. The result on the (E) datasets are in Table 9 while those for the (E, R) datasets are in Table 10.

### A.3  PERFORMANCE OF OTHER POTENTIAL SHORTCUTS

In our study, we show that PPR can achieve strong performance on most inductive KG datasets, indicating a shortcut. A natural question is whether this is true for just PPR, or if other shortcuts exist. Another bias discussed in KG literature is degree bias Shomer et al. (2023a). In their study,

Table 9: (E) Inductive Results for ULTRA Galkin et al. (2023)

| Metric | CoDEx-M | WN18RR | | HetioNet | |
|---|---|---|---|---|---|
| | Inference 1 | Inference 1 | Inference 2 | Inference 1 | Inference 2 |
| MRR | 30.2 | 64.7 | 21.4 | 57.9 | 72.7 |
| Hits@10 | 46.6 | 72.7 | 38.5 | 69.1 | 86.3 |

Table 10: (E) Inductive Results for ULTRA Galkin et al. (2023)

| Metric | FB15k-237 | | CoDEx-M | |
|---|---|---|---|---|
| | Inference 1 | Inference 2 | Inference 1 | Inference 2 |
| MRR | 45.7 | 38 | 30.4 | 73.7 |
| Hits@10 | 69.6 | 64.5 | 45.7 | 91.3 |

they show that that KG methods tend to perform better on entities with a higher degree. Specifically, they observe that what matters is the degree of the entity being predicted (they refer to this as the "tail degree"). Based on this, we study whether the tail degree can is a good predictor of performance.

We show that results on a number of inductive and transductive datasets in Table 11. We report Hits@10 for all results. We find that the tail degree often achieves a much lower performance than PPR. Specifically, for the datasets in Table 11, while the PPR performance is on average only 31% lower than the SOTA, the tail degree performance is 84% lower. This indicates the severity of the PPR shortcut is much more severe as compared to other known biases like the tail degree.

Table 11: Performance of Tail Degree Shomer et al. (2023a), PPR, and SOTA method. Performance is in terms of Hits@10. We further bold the larger of the PPR and tail degree performance.

| Task | Dataset | Tail Degree | PPR | SOTA |
|---|---|---|---|---|
| Inductive | WN v1 | 10.4 | **77.1** | 82.6 |
| | WN v2 | 5.1 | **74.4** | 83.6 |
| | WN v3 | 10.1 | **45.2** | 58.2 |
| | WN v4 | 2.2 | **67.3** | 74.5 |
| | FB v1 | 20.5 | **41.2** | 60.7 |
| | FB v2 | 22.7 | **47.6** | 70.4 |
| | FB v3 | 16.8 | **43.5** | 67.7 |
| | FB v4 | 16.1 | **38.4** | 66.8 |
| Transductive | WN18RR | 2.0 | **46.2** | 59.9 |
| | FB15k-237 | **6.0** | 2.7 | 66.6 |
| | DBPedia100k | 2.9 | **30.2** | 41.8 |

## B    DATASETS

### B.1    EXISTING DATASETS

We detail the statistics of all existing transductive and inductive datasets in Tables 12, 13, and 14, respectively. We further include the licenses for each in Table 15. Note that we omit YAGO3-10 Mahdisoltani et al. (2013) as Akrami et al. (2020) show that the dataset is dominated by two duplicate relations, making most triples trivial to classify. Also, we omit NELL-995 and any inductive datasets derived from it due to findings by Safavi & Koutra (2020) that show that most triples in the dataset are either too generic or meaningless.

### B.2    NEW DATASETS

We further detail the statistics of all the new datasets in Tables 16 and 17.

Table 12: Statistics for Transductive Datasets.

| Dataset | #Entities | #Relations | #Train | #Validation | #Test |
|---|---|---|---|---|---|
| FB15k-237 Toutanova & Chen (2015) | 14,541 | 237 | 272,115 | 17,535 | 20,466 |
| WN18RR Dettmers et al. (2018) | 40,943 | 11 | 86,835 | 3,034 | 3,134 |
| CoDEx-S Safavi & Koutra (2020) | 2,034 | 42 | 32,888 | 1,827 | 1,828 |
| CoDEx-M Safavi & Koutra (2020) | 17,050 | 51 | 185,584 | 10,310 | 10,311 |
| CoDEx-L Safavi & Koutra (2020) | 77,951 | 69 | 551,193 | 30,622 | 30,622 |
| HetioNet Himmelstein et al. (2017) | 45,158 | 24 | 2,025,177 | 112,510 | 112,510 |
| DBpedia100k Ding et al. (2018) | 99,604 | 470 | 597,572 | 50,000 | 50,000 |

Table 13: Statistics for (E) Inductive Datasets.

| Dataset | #Relations | Train Graph | | Validation Graph | | | Test Graph | | |
|---|---|---|---|---|---|---|---|---|---|
| | | #Entities | #Triples | #Entities | #Triples | #Valid | #Entities | #Triples | #Test |
| FB15k-237 v1 Teru et al. (2020) | 180 | 1,594 | 4,245 | 1,594 | 4,245 | 489 | 1,093 | 1,993 | 205 |
| FB15k-237 v2 Teru et al. (2020) | 200 | 2,608 | 9,739 | 2,608 | 9,739 | 1,166 | 1,660 | 4,145 | 478 |
| FB15k-237 v3 Teru et al. (2020) | 215 | 3,668 | 17,986 | 3,668 | 17,986 | 2,194 | 2,501 | 7,406 | 865 |
| FB15k-237 v4 Teru et al. (2020) | 219 | 4,707 | 27,203 | 4,707 | 27,203 | 3,352 | 3,051 | 11,714 | 1,424 |
| WN18RR v1 Teru et al. (2020) | 9 | 2,746 | 5,410 | 2,746 | 5,410 | 630 | 922 | 1,618 | 188 |
| WN18RR v2 Teru et al. (2020) | 10 | 6,954 | 15,262 | 6,954 | 15,262 | 1,838 | 2,757 | 4,011 | 441 |
| WN18RR v3 Teru et al. (2020) | 11 | 12,078 | 25,901 | 12,078 | 25,901 | 3,097 | 5,084 | 6,327 | 605 |
| WN18RR v4 Teru et al. (2020) | 9 | 3,861 | 7,940 | 3,861 | 7,940 | 934 | 7,084 | 12,334 | 1,429 |
| ILPC-S Galkin et al. (2022) | 48 | 10,230 | 78,616 | 6,653 | 20,960 | 2,908 | 6,653 | 20,960 | 2902 |
| ILPC-L Galkin et al. (2022) | 65 | 46,626 | 202,446 | 29,246 | 77,044 | 10,179 | 29,246 | 77,044 | 10,184 |

Table 14: Statistics for (E, R) Inductive Datasets.

| Dataset | Train Graph | | | Validation Graph | | | Test Graph | | |
|---|---|---|---|---|---|---|---|---|---|
| | #Entities | #Rels | #Triples | #Entities | #Rels | #Triples | #Valid | #Entities | #Rels | #Triples | #Test |
| FB-25 Lee et al. (2023) | 5,190 | 163 | 91,571 | 4,097 | 216 | 17,147 | 5,716 | 4,097 | 216 | 17,147 | 5,716 |
| FB-50 Lee et al. (2023) | 5,190 | 153 | 85,375 | 4,445 | 205 | 11,636 | 3,879 | 4,445 | 205 | 11,636 | 3,879 |
| FB-75 Lee et al. (2023) | 4,659 | 134 | 62,809 | 2,792 | 186 | 9,316 | 3,106 | 2,792 | 186 | 9,316 | 3,106 |
| FB-100 Lee et al. (2023) | 4,659 | 134 | 62,809 | 2,624 | 77 | 6,987 | 2,329 | 2,624 | 77 | 6,987 | 2,329 |
| WK-25 Lee et al. (2023) | 12,659 | 47 | 41,873 | 3,228 | 74 | 3,391 | 1,130 | 3,228 | 74 | 3,391 | 1,131 |
| WK-50 Lee et al. (2023) | 12,022 | 72 | 82,481 | 9,328 | 93 | 9,672 | 3,224 | 9,328 | 93 | 9,672 | 3,225 |
| WK-75 Lee et al. (2023) | 6,853 | 52 | 28,741 | 2,722 | 65 | 3,430 | 1,143 | 2,722 | 65 | 3,430 | 1,144 |
| WK-100 Lee et al. (2023) | 9,784 | 67 | 49,875 | 12,136 | 37 | 13,487 | 4,496 | 12,136 | 37 | 13,487 | 4,496 |

Table 15: Dataset Licenses.

| Datasets | License |
|---|---|
| FB15k-237 | CC-BY-4.0 |
| WN18RR | MIT License |
| CoDEx-S/M/L | MIT License |
| HetioNet | CC0 1.0 Universal |
| DBpedia100k | None |
| FB15k-237 v1/v2/v3/v4 | None |
| WN18RR v1/v2/v3/v4 | None |
| ILPC-S/L | MIT License |
| FB-25/50/75/100 | None |
| WK-25/50/75/100 | None |

# C PERSONALIZED PAGERANK (PPR)

In this section we provide additional detail on the formulation of PageRank and Personalized PageRank (PPR) Page et al. (1999). We further detail how PPR is calculated and used in our experiments.

For a graph $G = (V, E)$, PageRank Page et al. (1999) calculates an importance score for each node. The importance score for a node $v$ is determined by computing the probability of a random walk of

Table 16: Statistics for New **(E)** Inductive Datasets.

| Dataset | Graph | # Edges | # Entities | # Rels | # Valid/Test | $\Delta$SPD |
|---------|-------|---------|-----------|--------|--------------|------|
| CoDEx-M | Train | 76,960 | 8,362 | 47 | 8,552 | NA |
|         | Inference 1 | 69,073 | 8,003 | 40 | 7,674 | 0.24 |
| WN18RR | Train | 24,584 | 12,142 | 11 | 2,458 | NA |
|        | Inference 1 | 18,258 | 8,660 | 10 | 1,831 | 4.87 |
|        | Inference 2 | 5,838 | 2,975 | 8 | 572 | 1.47 |
| HetioNet | Train | 101,667 | 3,971 | 14 | 11,271 | NA |
|          | Inference 1 | 49,590 | 2,279 | 11 | 5,490 | 0.38 |
|          | Inference 2 | 37,927 | 2,455 | 12 | 4,187 | 0.19 |

Table 17: Statistics for New **(E, R)** Inductive Datasets.

| Dataset | Graph | # Edges | # Entities | # Rels | # Valid/Test | % New Rels | $\Delta$SPD |
|---------|-------|---------|-----------|--------|--------------|------------|------|
| FB15k-237 | Train | 45,597 | 2,864 | 104 | 5,062 | NA | NA |
|           | Inference 1 | 35,937 | 1,835 | 72 | 3,898 | 62.8% | 0.63 |
|           | Inference 2 | 51,693 | 2,606 | 143 | 5,735 | 27.1% | 0.34 |
| CoDex-M | Train | 29,634 | 4,038 | 36 | 3293 | NA | NA |
|         | Inference 1 | 70,137 | 7,938 | 39 | 7,794 | 9.9% | 0.24 |
|         | Inference 2 | 8,821 | 474 | 28 | 979 | 56.8% | 0.60 |

any length ending in that node. The rationale is that the nodes that are most likely to be visited, are most important to the underlying graph topology. The pagerank score for all nodes, $\mathrm{pr} \in \mathbb{R}^{|V|}$, can be computed by **(1)** randomly being the walk at any node with equal probability, **(2)** continuously applying random walks until we reach a stationary distribution, and **(3)** randomly teleporting back to the starting node with probability $\alpha$. Higher values of $\alpha$ help discourage the importance of longer walks. This can be computed from the following recurrence relation:

$$\mathrm{pr}^{(t+1)} = (1 - \alpha)W\mathrm{pr}^{(t)} + \alpha\mathrm{pr}^{(0)}, \tag{2}$$

where $W$ is the random walk matrix $D^{-1}A$ and $\mathrm{pr}^{(0)} = \mathbf{1}/|V|$. The pagerank score is equivalent to $t = \infty$.

Personalized PageRank (PPR) Page et al. (1999) generalizes this formulation by allowing us to customize which nodes we begin the random walk at, i.e., $\mathrm{pr}^{(0)}$. For example, we may only be interested in the PageRank score relative to a single node $s$. As such, the PageRank scores are *personalized* to the node $s$. We denote this quantity as $\mathrm{pr}_s$. This can be computed in the same manner as Eq. 2 by further modifying the initial probability vector $\mathrm{pr}^{(0)}$ to the relevant starting nodes:

$$\mathrm{pr}^{(0)} = \begin{cases} 1 & \text{when } i = s \\ 0 & \text{else} \end{cases}$$

where $i$ denotes entry $i$ in the vector. Note that PPR is not restricted to only one seed node, but can be personalized to a set of nodes $S$. PPR can be equivalently expressed as the weighted sum of random walks beginning from $s$:

$$\mathrm{pr}_s = \alpha \sum_{k=0}^{\infty} (1 - \alpha)^k W^k \mathrm{pr}^{(0)}. \tag{3}$$

In practice, directly computing the recurrence relation Eq. 2 or Eq. 3 is prohibitively expensive. Multiple approximation approaches have been proposed to more effectively compute the PPR score such as Andersen et al. (2006), which we employ in our paper.

In our experiments, we calculate the PPR of KGs, which further include a relation. However, **we completely ignore any relational information when computing the PPR**. This is done by: **(1)** Removing all edge types and **(2)** Assigning an edge weight of 1 to all edges. We further remove any directional information from the edges, thus transforming the graph to an undirected graph. The PPR

is then computed on this new undirected graph. We set $\alpha = 0.15$ in all our experiments. However, despite these modification, we show in Section 3 that PPR can still achieve strong performance during testing. This is due to a shortcut introduced during the dataset creation that allows for PPR to discriminate between positive and negative facts.

# D INDUCTIVE GENERATION EXPERIMENTS

We give more details on the experiments conducted in Section 3.3 and shown in Figures 4. Given the transductive dataset FB15k-237 Toutanova & Chen (2015), we generate a number of different inductive datasets using the common procedure used by Teru et al. (2020); Lee et al. (2023). We describe this procedure in detail in Section 2. Note that for simplicity, we only created datasets for the (E) inductive task. However, the same conclusions should hold for (E, R). Lastly, we only create one graph for inference as is done in previous work.

We generate the inductive datasets by modifying the following set of parameters:

- **# of Initial Train Entities**: This is the number of entities that are first assigned to the train graph.

- **# of Initial Inference Entities**: This is the number of entities that are first assigned to the inference graph.

- **Max Train Neighborhood Size**: We extract the 2-hop neighborhood for each of the initial train entities. To avoid exponential growth, we limit the number of entities selected in each hop to 50. For example, if node 1 has 30 1-hop neighbors and 120 2-hop neighbors, then $30 + 50 = 80$ entities are added through the expansion of node 1.

- **Max Inference Neighborhood Size**: This follows the same logic as for train but for inference.

The default neighborhood size is set to 50 for the inductive datasets created by Teru et al. (2020) and Lee et al. (2023). For Lee et al. (2023) they consider 10 and 20, initial entities for train and inference, respectively. This information is not reported for Teru et al. (2020).

To simulate the effect of the different parameter values, we create inductive datasets by using a number of different combinations of parameters. For each parameter, we modify it while holding the others constant. This allows us to explore the effect of just that variable. It allows us to avoid running an excessive amount of experiments. The default values are those used in Lee et al. (2023) and noted above. In total, there are 17 parameter configurations. For each parameter configuration, we create 3 different datasets through the use of 3 different random seeds. This results in a total of 51 inductive datasets. We then evaluate the performance of PPR on each inference graph and calculate the $\Delta$SPD. The results for those datasets with the same configuration are then averaged together. All possible configurations and their resulting mean statistics can be found in Table 18. These results are visualized in the main content in Figures 4.

Table 18: FB15k-237 - Inductive Generative Experiments Configurations. Results are over 3 Random Seeds.

| # Train Ents | # Inf. Ents | Max Train | Max Inf. | # Train Edges | # Test Edges | $\Delta$SPD | PPR Hits@10 |
|---|---|---|---|---|---|---|---|
| 10 | 20 | 10 | 50 | 7,599 | 72,688 | 1.14 | 0.166 |
| 10 | 20 | 15 | 50 | 18,518 | 49,057 | 1.31 | 0.180 |
| 10 | 20 | 25 | 50 | 43,077 | 36,054 | 1.46 | 0.213 |
| 10 | 20 | 50 | 50 | 91,843 | 18,370 | 1.71 | 0.309 |
| 10 | 20 | 100 | 50 | 129,782 | 15,179 | 1.72 | 0.235 |
| 10 | 20 | 50 | 10 | 91,843 | 4,304 | 2.15 | 0.400 |
| 10 | 20 | 50 | 25 | 91,843 | 13,374 | 1.79 | 0.319 |
| 10 | 20 | 50 | 50 | 91,843 | 18,370 | 1.71 | 0.309 |
| 10 | 20 | 50 | 100 | 91,843 | 21,512 | 1.71 | 0.310 |
| 10 | 20 | 50 | 50 | 91,843 | 18,370 | 1.71 | 0.309 |
| 20 | 20 | 50 | 50 | 149,635 | 8,258 | 2.38 | 0.317 |
| 40 | 20 | 50 | 50 | 210,109 | 973 | 3.89 | 0.408 |
| 10 | 10 | 50 | 50 | 91,843 | 11,177 | 1.6 | 0.325 |
| 10 | 20 | 50 | 50 | 91,843 | 18,370 | 1.71 | 0.309 |
| 10 | 40 | 50 | 50 | 91,843 | 36,353 | 1.81 | 0.266 |
| 10 | 80 | 50 | 50 | 91,843 | 47,419 | 1.83 | 0.254 |
| 10 | 160 | 50 | 50 | 91,843 | 55,233 | 1.93 | 0.259 |

# E    GRAPH PARTITIONING

In this section we give a basic overview of the graph partitioning problem. Graph partitioning seeks to partition a graph $G = (V, E)$ into $N$ subgraphs with disjoint entities Buluç et al. (2016). Formally this is formulated as the following where $V_i$ is the set of entities in the $i$-th partition.

$$V_1 \cup V_2 \cup \cdots \cup V_N = V, \tag{4}$$

$$V_1 \cap V_2 \cap \cdots \cap V_N = \emptyset. \tag{5}$$

However, an exact solution to above problem is infeasible in all but simple cases, as graph partitioning is known to be NP-Hard. Therefore, approximations are typically used to partition the graph.

Several popular approaches exist to solving this problem. The common thread between them is that they seek to extract partitions that contain many links between nodes of the same partition, but few between nodes of other partitions. As such, this allows us to discover distinct "communities" in the graph, that contain nodes that more often interact with one another. Spectral clustering Shi & Malik (2000) is a popular approach to graph partitioning that uses the Fiedler vector (i.e., the eigenvector of the 2nd smallest eigenvalue) to partition the graph. This is due to its connection the normalized cut of the graph. Another popular approach is the Louvain method Blondel et al. (2008). Louvain uses a specially designed objective, referred to as modularity, to identify partitions that are internally dense but feature few links between partitions.

# F    ADDITIONAL EXPERIMENTAL SETTINGS

**Training Settings**: Each model was trained on either a: Tesla V100 32Gb, NVIDIA RTX A6000 48Gb, NVIDIA RTX A5000 24Gb, or Quadro RTX 8000 48Gb. All models were implemented in PyTorch Paszke et al. (2019) and Torch-Geometric Fey & Lenssen (2019).

**Hyperparameters**: Due to the different types and ranges of hyperparameters used for each method, the exact hyperparameters and their ranges differ by method. For each method, we tried to follow the recommended ranges used by the authors. For InGram Lee et al. (2023) we tuned the learning rate in $\{1e^{-3}, 5e^{-4}\}$ and the number of entity layers in $\{2, 4\}$. For NodePiece Galkin et al. (2021) we tuned the learning rate from $\{1e^{-3}, 1e^{-4}\}$ and the loss margin from $\{15, 25\}$. For RED-GNN Zhang & Yao (2022), we tuned the learning rate from $\{5e^{-3}, 5e^{-4}\}$ and the dropout from $\{0.1, 0.3\}$. In their experiments, NBFNet Zhu et al. (2021) uses the same hyperparameters across all datasets. We therefore do the same. For Neural LP Yang et al. (2017) we also use the default set of hyperparameters that is shared across datasets.

**Evaluation**: For a given test triple $(s, r, o)$, we strive to predict both $s$ and $o$ individually. This is framed as a ranking problem where we want the probability of the correct entity (e.g., $o$) to rank higher than all possible negative entities. We can calculate the rank of the true entity over all negatives by the following where $p(\cdot)$ denotes the model probability for a triple:

$$\text{Rank(s, r, o)} = |\mathcal{V}| - \sum_{v \in \mathcal{V} \setminus \{o\}} \mathbf{1}\left(p(s, r, o) > p(s, r, v)\right). \tag{6}$$

We note that a lower rank, indicates better performance. In practice, some of the negative entities will be true triples observed during training. As such, following Bordes et al. (2013) we use the filtered setting and remove those entities from the ranking. Once we obtain the rank of the true entity (i.e., "how many negative entities does it have a higher probability than") we compute the Hits@K%. Hits@K% computes the percentage of sample where the following inequality holds true, Rank $\leq K$. We follow previous work Galkin et al. (2023); Zhang & Yao (2022); Lee et al. (2023) and evaluate against all possible negative entities.

# G    ADDITIONAL EXPERIMENTAL ANALYSIS

In this section we expound on the results shown in Section 5.

**(E) Results.** The results under the (E) setting are shown in Table 3. We can observe that NBFNet and RED-GNN can consistently outperform all other methods. This is consistent with theoretical

analysis provided by Huang et al. (2024) that show that the added expressiveness of these methods are vital for KGC. Furthermore, we can see that Neural LP and NodePiece tend to struggle. Both are consistent with previous results. For Neural LP, this likely stems from its limited ability to capture longer paths, as it must explicitly materialize them. NodePiece uses CompGCN Vashishth et al. (2019) as its backbone model. However, recent work has shown that standard GNN are ineffective for KGC, thus contributing to its poor performance Li et al. (2023). Lastly, we can see that while both versions of InGram struggle, DEq-InGram shows a noteworthy improvement over the original version. This suggests that the additional expressivity brought by double equivariance is beneficial.

**(E, R) Results.** The results under the (E) setting are shown in Table 4. Similar to the (E) setting, both RED-GNN and NBFNet rank highly across datasets. A similar observation can be made in regards to the poor performance of Neural LP and NodePiece. The poor results for InGram however are unexpected, as previous work Lee et al. (2023) shows that InGram excels in the (E, R) setting. Multiple reasons exist for this divergence. Most prominent is that Lee et al. (2023) only report results over 1 random seed while we report over 5. This may distort the results. It may also be that mitigating the shortcut has a stronger impact on InGram than other methods. We leave this exploration for future work. Interestingly however, DEq-InGram shows a large improvement over InGram. In fact, it is the most consistent of all method on the (E, R) task, ranking in the top 2 across all inference graphs. This again gives credence to the argument made by Gao et al. (2023) about the importance of double permutation- equivariant representations in inductive KG reasoning.

**Relative Performance on Old vs. New Datasets.** In Table 5 we display the change in performance of various methods on the old and new inductive datasets. Our analysis is limited to WN18RR (E) and FB15k-237 (E, R). For WN18RR (E), the older datasets constitute "WN (v1-v4)" introduced by Teru et al. (2020). For FB15k-237 (E, R) it's comprised of "FB (25-100)" created by Lee et al. (2023). The results are averaged across splits on the older datasets and across inference graphs for the newer datasets. For HetioNet (E), no prior datasets exist. For both versions of CoDEX, the older inductive datasets lack results, rendering us unable to make any comparison. For WN18RR (E) we observe a consistently large decrease in performance across all methods. The % difference ranges from 27-61% with a mean of 40.6%. This suggests that our new WN18RR datasets are much harder. For FB15k-237, the performance of most methods decrease, with a mean decrease of 9.5%. However, we do see a small increase in the performance of some methods. Nonetheless, this doesn't imply that these new datasets are "easier" for those methods for a few reasons. First, for the older datasets we use the performance reported by Lee et al. (2023). The results are only reported over 1 random seed while ours is over 5. This makes them difficult to compare, as the performance of just one random seed is unreliable. Second, the increase in performance for RED-GNN and NBFNet is small at $< 10\%$. Given that the older performance is only over one seed, this difference is not statistically significant. Lastly, for Neural LP on FB15k-237 (E, R), the standard deviation in performance is very high at 9.9 and 12.8, respectively on both inference graphs (see Table 4). As such, since the increase in performance relative to the older splits is only 3.8, this change is also not statistically significant.

# H ULTRA SETTINGS

When evaluating ULTRA Galkin et al. (2023) we use the 0-shot setting. By default, we use the checkpoint they provided that was trained on the transductive datasets: FB15k-237 Toutanova & Chen (2015), WN18RR Dettmers et al. (2018), CoDEx-M Safavi & Koutra (2020), and NELL-995 Xiong et al. (2017). However, since we create our own inductive splits from FB15k-237, CoDEx-M, and WN18RR, there is risk of test leakage. That is, some triples that may have been in the transductive training graph, may now be a test sample in one of our splits. This gives ULTRA an unfair advantage as it has already seen that triple.

To combat this issue, we train three different versions of ULTRA that omits the transductive dataset used to created that specific inductive dataset. This includes:

- *w/o FB15k-237*: Trained using WN18RR, CoDEx-M, and NELL-995.

- *w/o WN18RR*: Trained using FB15k-237, CoDEx-M, and NELL-995.

- *w/o CoDEx-M*: Trained using FB15k-237, WN18RR, and NELL-995.

We follow the same settings used to train the original checkpoints provided by the authors of UL-TRA. Lastly, since HetioNet is not one of the four datasets used, no additional model is needed.

The results are shown in Table 19, where the "W/o" column represents the results when removing the parent transductive dataset and "With" corresponds to the original pre-trained checkpoint provided by the authors of ULTRA Galkin et al. (2023). In practice, we observe a small but noteworthy decrease in performance when removing the offending transductive dataset from the pre-trained model.

Table 19: Performance (**Hits@10**) of ULTRA when pre-trained and w/o the parent transductive dataset. Note that this doesn't apply to HetioNet.

| Dataset | Inference Graph | W/o | With | % Difference |
|---|---|---|---|---|
| CoDEx-M (E) | 1 | 46.6 | 50.2 | -7.2% |
| WN18RR (E) | 1 | 71.9 | 72.7 | +1.1% |
| | 2 | 38.8 | 38.5 | -0.8% |
| FB15k-237 (E, R) | 1 | 69.6 | 72.1 | -4.5% |
| | 2 | 64.5 | 64.8 | -0.5% |
| CoDEx-M (E, R) | 1 | 45.7 | 50.4 | -9.3% |
| | 2 | 91.1 | 91.3 | -0.2% |

# I  ANALYSIS OF THE WIKITOPICS AND PEDIATYPES DATASETS

We further include an analysis of the WikiTopics and PediaTypes inductive datasets introduced by Gao et al. (2023). For these datasets, the train and inference graphs contain completely separate entities and relations. For each dataset we record the **(a)** Neural SOTA performance (taken from Gao et al. (2023), **(b)** PPR performance, **(c)** % difference in performance, and **(d)** $\Delta$SPD. Note that in their original study, Gao et al. (2023) use 50 random negatives during evaluation, instead of all negative entities used in ours. To account for this, we compute the PPR performance using both evaluation settings. The results for the PediaTypes and WikiTopics datasets are in Tables 20 and 21, respectively.

From the results we can make multiple observations. First, PPR performs well under both settings. Under the all negatives setting, the mean PPR performance is 32.5% and 38.3% on PediaTypes and WikiTopics, respectively. Second, under 50 negatives, there is little difference in the PPR and Neural performance. Specifically, the % difference in performance is 0.4% and 14.2%, respectively, on PediaTypes and WikiTopics. This suggests that neural methods can only slightly outperform PPR under this setting. Lastly, the $\Delta$SPD is high on both sets of datasets. This further corroborates our findings in Section 3.

This analysis shows that the WikiTopics and PediaTopics datasets introduced by Gao et al. (2023) fall prey to the same shortcut found in other inductive datasets studied in Section 3.

Table 20: Performance on PediaTypes Gao et al. (2023) datasets. Note that since the original paper used 50 random negative samples during evaluation, we thus include this results.

| Dataset | 50 Negatives | | | All Negatives | |
|---|---|---|---|---|---|
| | Neural Hits@10 | PPR Hits@10 | % Difference | PPR Hits@10 | $\Delta$SPD |
| EN-FR | 95.4% | 96.0% | -0.6% | 44.1% | 2.7 |
| FR-EN | 90.6% | 96.2% | -5.8% | 48.1% | 2.3 |
| EN-DE | 97.7% | 93.9% | 4.0% | 34.7% | 2.1 |
| DE-EN | 95.0% | 90.6% | 4.9% | 20.8% | 1.6 |
| DB-WD | 86.6% | 76.4% | 13.4% | 23.5% | 0.6 |
| WD-DB | 91.5% | 92.0% | -0.5% | 42.9% | 2.5 |
| DB-YG | 71.5% | 76.9% | -7.0% | 15.2% | 2.7 |
| YG-DB | 80.5% | 85.1% | -5.4% | 30.7% | 3.5 |

Table 21: Performance on WikiTopics Gao et al. (2023) datasets. Note that since the original paper used 50 random negative samples during evaluation, we thus include this results.

| Dataset | 50 Negatives | | | All Negatives | |
|---------|--------------|----------|--------------|---------------|------|
| | Neural Hits@10 | PPR Hits@10 | % Difference | PPR Hits@10 | $\triangle$SPD |
| Art | 92.3% | 79.3% | 16.4% | 14.7% | 1.1 |
| Award | 93.6% | 95.8% | -2.3% | 24.2% | 3.4 |
| Edu | 97.0% | 72.4% | 34.0% | 20.8% | 2.7 |
| Health | 98.9% | 91.5% | 8.1% | 59.6% | 3.3 |
| Infra | 97.8% | 97.2% | 0.6% | 61.4% | 5.4 |
| Sci | 96.7% | 84.3% | 14.7% | 25.5% | 4.4 |
| Sport | 81.2% | 82.9% | -2.1% | 14.4% | 0.3 |
| Tax | 93.7% | 64.1% | 46.1% | 29.0% | 0.8 |
| Loc | N/A | 96.5% | N/A | 74.1% | 1.4 |
| Org | N/A | 94.0% | N/A | 8.9% | 2.8 |
| People | N/A | 94.9% | N/A | 64.6% | 2.7 |

## J  IMPLEMENTATION

All code used for experiments in this study can be found in the following anonymized repository – https://anonymous.4open.science/r/KG-PPR-ICLR-D0B7. Please see the README in the repository for more information on how to run the different experiments.

## K  LIMITATIONS

One potential limitation of our study is that the new inductive datasets are still being created from existing transductive datasets. While our method can help create better and more realistic inductive scenarios, this is limiting as we are still reliant on good quality transductive datasets. Future work can focus on creating inductive KGC datasets directly from existing large-scale knowledge graphs, thereby bypassing the need to sample from existing transductive datasets.

## L  IMPACT STATEMENT

Our method contributed positively to the field of knowledge graph reasoning. By introducing newer and better inductive datasets, we better align the task of KGC to it's real-world applications. This is essential, because if we want to perform inductive KGC in real-world tasks, we have to be certain that the methods actually work well. As such, the new datasets can serve as a stronger barometer of actual performance on inductive KGC and will help spur the development of newer and more effective KGC techniques. Since KGC has applications in many different fields including question answering, biology, and recommender systems, there is a strong need for methods that can perform well. We have also carefully considered the broader impact of our work from different perspectives and find that there is no apparent risk.

