# OpenReview forum: "Towards Better Benchmark Datasets for Inductive Knowledge Graph Completion"
_ICLR.cc/2025/Conference — Submitted to ICLR 2025_

### Official Review · Reviewer_JoYY · 2024-10-29

**Soundness:** 3
**Presentation:** 3
**Contribution:** 2
**Rating:** 6
**Confidence:** 4

**Summary:**

This paper studies the datasets for inductive KGC. First, it is observed that PPR can have hear SOTA performance on most inductive datasets. Then, some analytical experiments are conducted to study the correlation between PPR and shortest path distance. Based on the observations, the authors propose a new split method through graph participation algorithms.

**Strengths:**

1. The logic in this paper is smooth and easy to capture.
2. The analysis of the relationship between PPR, SDP, and KGC performance is studied in detail.
3. The results of the split generated by the graph partition method are consistent with the detailed analysis.

**Weaknesses:**

1. Even though the evaluation of PPR and SPD on current inductive benchmarks reals some problems and sounds reasonable, the match of these problems with realistic demands is not fully discussed. In particular, there lack of discussion and evaluation of datasets in real-world inductive settings. At the beginning of the introduction, KGC applications include drug discovery, personalized medicine, and recommendations. Is it possible to evaluate some cases of new drugs, new patients, and cold start recommendations? I want to know whether phenomena like the correlation between PPR and SPD exist in these applications.

2. Similar to problem 1, I think the main purpose of recreating the inductive dataset is to match it with its transductive counterparts. In my mind, the transductive datasets also have unrealistic problems.

3. Some details of the graph partition method in data creation are not clear. In particular, I care about how the graph partition method can guarantee that the training and inference datasets have different sets of entities or relations. Maybe I missed some important content, but I failed to find this point in both the main content and the appendix.

4. Another commonly used inductive dataset NELL is not studied in this paper.

**Questions:**

1. Is it possible to evaluate some cases of new drugs, new patients, and cold start recommendations?

2. Whether phenomena like the correlation between PPR and SPD exist in these applications?

3. How to guarantee the sets of entities or relations are different by graph partitioning?

---

> ### Author Response · Authors · 2024-11-19
> **Response to Reviewer JoYY (1/2)**
>
> We appreciate your help in improving our paper through you comments and questions. We respond to them below.
>
> > In particular, there is a lack of discussion and evaluation of datasets in real-world inductive settings.
> >
> > Is it possible to evaluate some cases of new drugs, new patients, and cold start recommendations? I want to know whether phenomena like the correlation between PPR and SPD exist in these applications.
>
> We appreciate the comment and would like to clarify a few details.
>
> **(1)** We agree that there should be a focus on real-world KGs. In fact, the KGs used in our study span a number of different domains. This includes:
> - **General/Encyclopedic**: This includes FB15k-237, CoDEx, and DBPedia100k.
> - **Drug Discovery**: HetioNet
> - **Lexical/Linguistic**: WN18RR
>
> Our findings are consistent across each. We are thus confident that our analysis should apply to KGs from different domains.
>
> **(2)** Our findings aren't related to the domain of the KG but due to how inductive datasets are constructed. In our study we find that regardless of the original KG, the older strategy for conducting inductive KG splits leads to this shortcut in the inference graph. This is because the shortcut is only related to the topology of the inference graph. This is why the PPR performance always increases significantly from transductive to inductive KGs.
>
> Additionally, **this is not something that is a concern in real-world datasets, as we don't explicitly sample a train and inference graph from a larger KG**. In real-world situations, we are not given the train and inference KGs a priori. Rather, we simply seek to learn on one KG and transfer that knowledge to another. However, as shown in our paper, the older dataset strategy constructs datasets that exhibit unique patterns not found in other real-world KGs. **It therefore doesn't reflect actual inductive KGC as we would not expect for the inference graph to have such properties**.
>
> We show in Section 4 that our proposed dataset construction strategy can indeed create inductive datasets that aren't unnecessarily modified and thus correspond to real-world KGs.
>
> > I think the main purpose of recreating the inductive dataset is to match it with its transductive counterparts. In my mind, the transductive datasets also have unrealistic problems.
>
> Thank you for your comment. We agree that we want to sample the train and inference graphs such that their graph properties are similar to the transductive dataset. However we are unsure what kind of "unrealistic problems" you're referring to for transductive KGs. We believe that the existing transductive KGs are a good gauge of real-world KGC performance. This is due to two reasons:
> 1. These datasets often reflect real-world KGs. For example FB15k-237 is a subset of FreeBase, a prominent KG [1].
> 2. We don't observe the PPR shortcut on most transductive KGs.
>
> If you would further expound on your thoughts, we would be glad to try to answer them.
>
> [1] Bollacker, Kurt, et al. "Freebase: a collaboratively created graph database for structuring human knowledge." Proceedings of the 2008 ACM SIGMOD international conference on Management of data. 2008.
>
> > Another commonly used inductive dataset NELL is not studied in this paper.
>
> We omit NELL-995 from our study due to quality concerns raised by [1]. We include a note of this in our original paper (see lines 804-806 in the revised version). Specifically, [1] shows that many of the triples used in the dataset are either too generic or meaningless. For example, [1] includes examples of two triples present in NELL-995 which are trivially wrong. We show them below:
> - "(politician:jobs, worksfor, county:god)"
> - "(person:buddha001, parentofperson, person:jesus)"
>
> For more context, NELL-995 is derived from the Never Ending Language Learner (NELL) KG [2]. NELL automatically extracts facts from the internet. Each fact is further assigned a confidence regarding its truthfulness. E.g., a 90% implies the fact has a 90% chance of it being true. [3] used a version of NELL to construct NELL-995. However, **they don't take into account the fact confidence when extracting facts for NELL-995**. As such, it is likely that many facts are unlikely to be true. This is likely the cause for the observations made by [1].
>
> Due to these observations, we don't think NELL-995 is an accurate reflection of real-world KGs, as many of its facts may be false.
>
> [1] Safavi, Tara, and Danai Koutra. "CoDEx: A Comprehensive Knowledge Graph Completion Benchmark." EMNLP. 2020.
> [2] Tom Mitchell, et al. 2018. "Never-ending learning". CACM, 2018.
> [3] Xiong, et al. "DeepPath: A Reinforcement Learning Method for Knowledge Graph Reasoning." EMNLP, 2017.

---

> > ### Author Response · Authors · 2024-11-19
> > **Response to Reviewer JoYY (2/2)**
> >
> > > How to guarantee the sets of entities or relations are different by graph partitioning?
> >
> > We apologize for any confusion.
> >
> > By definition, graph partitioning results in disjoint set of entities (i.e., nodes) among the different partitions [1]. Specifically, assume we are given a graph $G=(V, E)$ where $V$ and $E$ are the set of entities and edges, respectively. The goal of graph partitioning is to obtain disjoint sets of entities, $V\_i \in V$ such that if we have $N$ partitions:
> > $$
> > V\_1 \cup V\_2 \cup \cdots \cup V\_N = V
> > $$
> > $$
> > V\_1 \cap V\_2 \cap \cdots \cap V\_N = \emptyset
> > $$
> > The second constraint above must result in the partitions containing no overlapping entities. To enhance the clarity of our paper, we include this discussion in our revised paper in Appendix E.
> >
> > Furthermore, **we are actually not trying to sample disjoint sets of relations**. Let's denote the relations in the train and inference graph as $R\_{train}$ and $R\_{inf}$. For both types of datasets we want the relations to be as the following:
> > - **(E) datasets**: All inference relations must be seen in the training graph. I.e., $R\_{inf} \subseteq R\_{train}$. This is done by sampling two graphs, train and inference, and removing those edges from inference that contain relations not seen in training. In practice, this is very easy and we always have to remove very few edges from inference (usually 0 but never more than a handful of edges).
> > - **(E, R) datasets**: There is no constraint that the relations in inference must be seen during training. We simply choose two graphs for training and inference, without modification.
> >
> > [1] Buluç, Aydın, et al. Recent advances in graph partitioning. Springer International Publishing, 2016.

---

> ### Author Response · Authors · 2024-11-25
>
> Dear Reviewer JoYY,
>
> Thank you for taking the time to review our paper!
>
> As the discussion deadline is rapidly approaching, we are eager to hear your response on our rebuttal as soon as possible. We hope that all aspects of our response addresses your concerns and await additional questions.
>
> Regards,
>
> Authors

---

> > ### Comment · Reviewer_JoYY · 2024-11-25
> > **Thank you for your response**
> >
> > I appreciate the authors' efforts in responding to my reviewing comments. The score has been updated.

---

> > > ### Author Response · Authors · 2024-11-25
> > >
> > > We are happy that our response clarified your concerns. Please let us know if you have any other questions. We'll be happy to answer them.

---

### Official Review · Reviewer_mSV8 · 2024-11-01

**Soundness:** 2
**Presentation:** 3
**Contribution:** 3
**Rating:** 6
**Confidence:** 4

**Summary:**

This paper identifies a shortcut in current inductive KGC datasets, where simple methods like Personalized PageRank can achieve strong performance without using relational information. The authors propose a new dataset construction strategy to eliminate this shortcut and benchmark popular KGC methods on these datasets.

**Strengths:**

1. This work provides a clearer understanding of the capabilities and challenges in inductive KGC and the benchmark construction.

2. The manuscript is well-organized and easy to follow.

3. Experiments have verified the effectiveness of the proposed benchmarks.

**Weaknesses:**

1. The technical contribution is kind of limited for a long paper. The applied graph partitioning strategy is straightforward and the research challenge in this paper is not significant.

2. There is a confusing point in the benchmark construction. Why the differences in SPD are beneficial from graph partitioning sampling? Since “the different in mean SPD” (also a typo in line 237) causes this shortcut, why not improve the negative sampling strategy directly? For example, select entities having a much shorter distance from the query entity as negative samples. It could be independent of the graph structure.

3. In my opinion, the shortcut proposed in this paper is mainly caused by the negative-sampled evaluation (distinguishing the positive entity from a fixed number of negative ones). Recent studies, including RED-GNN and NBFNet, already employ the full evaluation (finding the positive entity from the entire entity set). This might be the reason for the superior performance of the two models as well as ULTRA. From this point, have the authors compared the performance differences between the two evaluation settings on new benchmarks?

**Questions:**

Please refer to Weaknesses.

---

> ### Author Response · Authors · 2024-11-19
> **Response to Reviewer mSV8**
>
> Thank you for taking the time to review our paper. We reply to your questions below.
>
> > The applied graph partitioning strategy is straightforward and the research challenge in this paper is not significant.
>
> We politely disagree with the reviewer as to the significance of our paper. Specifically, while we agree that the graph partitioning strategy is *simple*, we would like to note that it isn't the main contribution of our paper. Rather, **our contribution is in the discovery, diagnosis, and correction of this shortcut**. Furthermore, we argue that creating better and more realistic benchmark datasets is extremely important in fostering better understanding of real-world performance of different methods. We summarize our main contributions can be summarized as:
>
> 1) **Discovering the widespread prevalence of the shortcut on inductive KGC datasets**: We are the first to discover that this shortcut exists for inductive KGC datasets. This is important, as we show that **it afflicts essentially all such datasets** and results in an inflated test performance.
> 2) **Understanding the root cause of this shortcut**: We further seek to understand what causes this shortcut. We find that it is due to how previous datasets are constructed. This analysis is shown by **(a)** analyzing existing datasets (Sections 3.1-3.2) **(b)** generating our own datasets (Section 3.2 observation 3). These analyses empirically show that  difference in SPD distribution between positives and negatives leads to this shortcut. Furthermore, it is very difficult to mitigate this shortcut using the original construction strategy as it often leads to either the train or inference graph being very small.
> 3) **Introducing new datasets that mitigate the shortcut**: Now that we understand the cause, we are thus able to introduce a simple but effective new method for creating datasets that is borne from first principles. We show in Section 4 that our method can dramatically reduce the prevalence of this shortcut. **By removing the shortcut, the new datasets better reflect real-world applications of KGC and thus allow a better reflection of model performance than the older datasets**.
>
> Lastly, we'd like to note that the primary track/area of our paper is the **datasets+benchmarks** track. Therefore, **the focus of our work isn't to introduce new model methodologies, but to create better and more realistic benchmark datasets**. This is vital, because if we seek to truly understand the performance of different models and how they generalize to real-world applications, we must have effective and realistic benchmark datasets that reflect those situations.
>
> > In my opinion, the shortcut proposed in this paper is mainly caused by the negative-sampled evaluation
> >
> >  Recent studies, including RED-GNN and NBFNet, already employ the full evaluation (finding the positive entity from the entire entity set)
> >
> >  have the authors compared the performance differences between the two evaluation settings on new benchmarks?
>
> We agree with the reviewer, and in fact, **we actually do use the full evaluation setting and evaluate against all possible negative entities**. This setting is used **for all the results throughout our paper**. As such, the negative sampling strategy cannot be the cause of this shortcut.
>
> We choose to evaluate against all negative entities as we believe it reflects a more realistic scenario. In real-world situations we would in fact have to test against all entities, as we don't know which entity is correct ahead of time. Furthermore, some previous papers only use 50 random negatives for better efficiency, which while understandable, is not realistic in real-world settings where we don't have this ability.
>
> We have revised our paper to more clearly state which evaluation setting we are using. We revised our paper on line 955 (highlighted red) to include that. Specifically we include:
>
> > *We follow previous work Galkin et al. (2023); Zhang & Yao (2022); Lee et al.(2023) and evaluate against all possible negative entities.*
>
> We apologize for any confusion on the evaluation strategy.
>
> > why not improve the negative sampling strategy directly? For example, select entities having a much shorter distance from the query entity as negative samples. It could be independent of the graph structure
>
> As noted in the previous comment, we evaluate against all possible negative entities. As such, there is no sampling to be done, as all entities regardless of their distance are included in the ranking.

---

> ### Author Response · Authors · 2024-11-25
> **Reminder about Rebuttal**
>
> Dear Reviewer mSV8,
>
> Thank you for taking the time to review our paper!
>
> As the discussion deadline is rapidly approaching, we are eager to hear your response on our rebuttal as soon as possible. We hope that all aspects of our response addresses your concerns and await additional questions.
>
> Regards,
>
> Authors

---

> > ### Comment · Reviewer_mSV8 · 2024-11-25
> > **Official Response by Reviewer mSV8**
> >
> > Thanks to the authors for their detailed responses. I have raised my score.

---

> > > ### Author Response · Authors · 2024-11-25
> > >
> > > We are happy that our response clarified your concerns. Please let us know if you have any other questions. We'll be happy to answer them.

---

### Official Review · Reviewer_wYLD · 2024-11-04

**Soundness:** 3
**Presentation:** 3
**Contribution:** 3
**Rating:** 8
**Confidence:** 3

**Summary:**

Inductive link prediction aims to predict missing links in test sets where entities, relations, or both have not been encountered during training. Numerous datasets have been developed as benchmarks for evaluating inductive link prediction methods. This paper identifies a flaw in existing inductive link prediction datasets: a simple, non-trainable method called personalized PageRank (PPR) can achieve performance comparable to complex machine learning approaches. This is largely because current datasets are structured such that the shortest path distance (SPD) between entities in positive test samples is considerably shorter than in negative samples. Consequently, a non-trainable algorithm like PPR, which disregards relation information, can still rank positive samples above negative ones, yielding high performance.
To address this issue, the authors propose new datasets for inductive link prediction, carefully partitioned from transductive datasets to prevent the PPR shortcut. The new benchmark reveals that models perform worse on these updated datasets compared to older ones, highlighting the need for more robust evaluation standards in inductive link prediction.

**Strengths:**

-- Numerous machine learning and AI approaches have been introduced in the literature. To demonstrate the effectiveness of new models compared to older ones, several standard datasets have been established for evaluation and conclusion. However, issues with these datasets can lead to misleading conclusions within the community about model performance. Therefore, the paper's original motivation and the problem it addresses are both significant and valuable.

– Overall, the paper is written well and easy to follow and highlight different aspects.

– The results are consistent with what claimed, i.e., the performance of existing models on new inductive datasets are lower than older ones and also PPR obtain lower results on new datasets.

**Weaknesses:**

– Some part of the paper requires a better and more detailed explanation, e.g., the description related to page rank which is important to better understand the issue.

– that the deltaSPD of train and inference should be close is understandable. But it is not very clear why this partitioning solve the main raised issue.

–  the analysis of the tables should be more detailed and cover interpretation of various observations.

**Questions:**

Older datasets have simple patterns than new datasets. Why performance of NBFNet improves on new datasets while these datasets are more challenging?

Why the performance difference of PPR and NBFNet on WN18RR dataset in both old and new versions are the same, but this is not the case for FB237?

---

> ### Author Response · Authors · 2024-11-19
> **Response to Reviewer wYLD (1/N)**
>
> We appreciate your help in improving our paper through you comments and questions. We respond to them below.
>
> > Some part of the paper requires a better and more detailed explanation, e.g., the description related to page rank
>
> Thank you for your feedback, we strive to make our paper as readable as possible.
>
> We've included a more detailed description of pagerank and how it's used in our experiments in Appendix C. For clarity, we've also included the new portion below.
>
> > For a graph $G=(V, E)$, PageRank Page et al. (1999) calculates an importance score for each node. The importance score for a node $v$ is determined by computing the probability of a random walk of any length ending in that node. The rationale is that the nodes that are most likely to be visited, are most important to the underlying graph topology. The pagerank score for all nodes,  $\text{pr} \in \mathbb{R}^{\lvert V \rvert}$, can be computed by **(1)** randomly being the walk at any node with equal probability,  **(2)** continuously applying random walks until we reach a stationary distribution, and **(3)** randomly teleporting back to the starting node with probability $\alpha$. Higher values of $\alpha$ help discourage the importance of longer walks. This can be computed from the following recurrence relation:
> 	\begin{equation}
>     	\text{pr}^{(t+1)} = (1 - \alpha) W  \text{pr}^{(t)} + \alpha \text{pr}^{(0)},      \text{(2)}
> 	\end{equation}
> 	where $W$ is the random walk matrix $D^{-1} A$ and $\text{pr}^{(0)} = \mathbf{1} / \lvert V \rvert$. The pagerank score is equivalent to $t=\infty$.
> >
> > Personalized PageRank (PPR) Page et al. (1999) generalizes this formulation by allowing us to customize which nodes we begin the random walk at, i.e., $\text{pr}^{(0)}$. For example, we may only be interested in the PageRank score relative to a single node $s$. As such, the PageRank scores are *personalized* to the node $s$. We denote this quantity as $\text{pr}\_{s}$. This can be computed in the same manner as Eq. (2) by further modifying the initial probability vector $\text{pr}^{(0)}$ to the relevant starting nodes:
> 	\begin{equation} \nonumber
>     	\text{pr}^{(0)} =
>         	\begin{cases}
>           	1 & \text{when i = s } \\\\
>           	0 & \text{else} \\
>         	\end{cases}
> 	\end{equation}
> 	where $i$ denotes entry $i$ in the vector. Note that PPR is not restricted to only one seed node, but can be personalized to a set of nodes $S$. PPR can be equivalently expressed as the weighted sum of random walks beginning from $s$:
> 	\begin{equation}
>     	\text{pr}\_{s} = \alpha \sum\_{k=0}^{\infty} (1-\alpha)^k W^k \text{pr}^{(0)}.  \text{(3)}
> 	\end{equation}
> > In practice, directly computing the recurrence relation Eq. (2) or (3) is prohibitively expensive. Multiple approximation approaches have been proposed to more effectively compute the PPR score such as  Andersen et al. (2006), which we employ in our paper.
> >
> >
> > In our experiments, we calculate the PPR of KGs, which further include a relation. However, **we completely ignore any relational information when computing the PPR scores**. This is done by: **(1)** Removing all edge types and **(2)** Assigning an edge weight of 1 to all edges. We further remove any directional information from the edges, thus transforming the graph to an undirected graph. The PPR is then computed on this new undirected graph. We set $\alpha=0.15$ in all our experiments. However, despite these modifications, we show in Section 3 that PPR can still achieve strong performance during testing. This is due to a shortcut introduced during the dataset creation that allows for PPR to discriminate between positive and negative facts.
>
> Please let us know if there is any other part of the paper that you think is unclear and requires additional explanation.

---

> > ### Author Response · Authors · 2024-11-19
> > **Response to Reviewer wYLD (2/N)**
> >
> > > it is not very clear why this partitioning solve the main raised issue
> >
> > We argue that it's due to the inherent goal of graph partitioning algorithms, which attempts to extract partitions (or communities) from the graph that are distinct from one another. This limits any changes to the underlying graph topology, preserving the characteristics of the original graph. In our case, we want to sample train and inference graphs ($\mathcal{G}\_{\text{train}}$, $\mathcal{G}\_{\text{inf}}$) from a transductive dataset ($\mathcal{G}$) such that we minimize any change in the $\Delta \text{SPD}$. Formally, this is expressed the following:
> > $$
> > \Delta\text{SPD}(\mathcal{G}\_{\text{train}}) \approx \Delta\text{SPD}(\mathcal{G}_{\text{inf}}) \approx \Delta\text{SPD}(\mathcal{G}).
> > $$
> >
> > *But why should graph partitioning help us achieve this goal?* We believe that this is due to the objective that graph partitioning attempts to solve. Specifically,  graph partitoning attempts to extract partitions (or communities) from the graph that are distinct from one another. A common way of achieving this goal is by **minimizing the number of edges that link nodes in different partitions** (i.e., spectral clustering). By doing so, we can extract partitions (i.e., subgraphs) that alter the graph topology as little as possible.
> >
> > We given an example of this idea in Figure 5 in our paper, where even though the edges connecting the two partitions are removed, **the relationship between nodes in the same partition is essentially unchanged**. As such, we expect that the $\Delta \text{SPD}$ of the nodes in the resulting train and inference graphs (i.e., partitions) should also be relatively unchanged. We further verify this in Section 4.2, where we demonstrate that the change in $\Delta \text{SPD}$ is minor when using graph partitioning. This is not true for the older construction method where the difference is large. **These results demonstrate that graph partitioning is able to preserve the essential graph properties when sampling the train and inference graphs, thus not introducing any new shortcut.**
> >
> >
> > > the analysis of the tables should be more detailed and cover interpretation of various observations.
> >
> > We have included additional discussion of the results in the revised section of our paper. This is in Appendix G of our paper (we omit it from the main text due to page limitations).
> >
> > We further include this discussion below.
> >
> > > In this section we expound on the results shown in Section 5.
> > >
> > > **(E) Results**. The results under the (E) setting are shown in Table 3. We can observe that NBFNet and RED-GNN can consistently outperform all other methods. This is consistent with theoretical analysis provided by Huang et al. (2024) that show that the added expressiveness of these methods are vital for KGC. Furthermore, we can see that Neural LP and NodePiece tend to struggle. Both are consistent with previous results. For Neural LP, this likely stems from its limited ability to capture longer paths, as it must explicitly materialize them. NodePiece uses CompGCN Vashishth et al. (2019) as its backbone model. However, recent work has shown that standard GNN are ineffective for KGC, thus contributing to its poor performance Li et al. (2023). Lastly, we can see that while both versions of InGram struggle, DEq-InGram shows a noteworthy improvement over the original version. This suggests that the additional expressivity brought by double equivariance is beneficial.
> > >
> > > **(E, R) Results**. The results under the (E) setting are shown in Table 4. Similar to the (E) setting, both RED-GNN and NBFNet rank highly across datasets. A similar observation can be made in regards to the poor performance of Neural LP and NodePiece. The poor results for InGram however are unexpected, as previous work Lee et al. (2023) shows that InGram excels in the (E, R) setting. Multiple reasons exist for this divergence. Most prominent is that Lee et al. (2023) only report results over 1 random seed while we report over 5. This may distort the results. It may also be that mitigating the shortcut has a stronger impact on InGram than other methods. We leave this exploration for future work. Interestingly however, DEq-InGram shows a large improvement over InGram. In fact, it is the most consistent of all methods on the (E, R) task, ranking in the top 2 across all inference graphs. This again gives credence to the argument made by Gao et al. (2023) about the importance of double permutation- equivariant representations in inductive KG reasoning.
> >
> > *Continued in next comment*

---

> > > ### Author Response · Authors · 2024-11-19
> > > **Response to Reviewer wYLD (3/N)**
> > >
> > > > **Relative Performance on Old vs. New Datasets.** In Table 5 we display the change in performance of various methods on the old and new inductive datasets. Our analysis is limited to WN18RR (E) and FB15k-237 (E, R). For WN18RR (E), the older datasets constitute “WN (v1-v4)” introduced by Teru et al. (2020). For FB15k-237 (E, R) it’s comprised of “FB (25-100)” created by Lee et al. (2023). The results are averaged across splits on the older datasets and across inference graphs for the newer datasets. For HetioNet (E), no prior datasets exist. For both versions of CoDEX, the older inductive datasets lack results, rendering us unable to make any comparison. For WN18RR (E) we observe a consistently large decrease in performance across all methods. The % difference ranges from 27-61% with a mean of 40.6%. This suggests that our new WN18RR datasets are much harder. For FB15k-237, the performance of most methods decrease, with a mean decrease of 9.5%. However, we do see a small increase in the performance of some methods. Nonetheless, this doesn’t imply that these new datasets are “easier” for those methods for a few reasons. First, for the older datasets we use the performance reported by Lee et al. (2023). The results are only reported over 1 random seed while ours is over 5. This makes them difficult to compare, as the performance of just one random seed is unreliable. Second, the increase in performance for RED-GNN and NBFNet is small at < 10%. Given that the older performance is only over one seed, this difference is not statistically significant. Lastly, for Neural LP on FB15k-237 (E, R), the standard deviation in performance is very high at 9.9 and 12.8, respectively on both inference graphs (see Table 4). As such, since the increase in performance relative to the older splits is only 3.8, this change is also not statistically significant.
> > >
> > > Please let us know if you have any other questions about the results.
> > >
> > > > Older datasets have simple patterns than new datasets. Why does NBFNet improve on new datasets while these datasets are more challenging?
> > >
> > > This is a good question. From our analysis, we find that the change in performance of NBFNet is dataset dependent.
> > >
> > > We compare the change in performance for all methods on WN18RR (E) and FB15k-237 (E, R) in Table 5 in our paper. We restrict to those two datasets, as they are the only two that have relevant prior datasets with sufficient results. We show the change in performance of NBFNet below. Note that for the older datasets we take the mean across splits, while for the newer ones we take the mean across the inference graphs.
> > >
> > > **NBFNet Performance**:
> > > | Dataset | Old Performance | New Performance | % Change |
> > > | ------- | --------------  | ---------------------- | -------- |
> > > | WN18RR (E) | 72.2 | 52.5 | -27.4% |
> > > | FB15k-237 (E, R) | 24.8 | 26.9 | +8.3% |
> > >
> > > We can observe that the change is dataset dependent. On WN18RR (E) the performance drastically decreases. However, for FB15k-237 (E, R) it modestly improves. Upon examination, we believe that there are multiple explanations that may account for this:
> > > - The results on the older versions of FB15k-237 (E, R) are only over 1 random seed. Note that they are taken from [1]. Conversely, our results are averaged over 5 seeds. This is not true for the older versions of WN18RR (E) where both sets of results are over 5 seeds. This is problematic, as the result on only one random seed can be misleading due to variance. Given the small gap in performance between the old and new datasets, the fact that they only report performance over one seed **suggests that the change in performance is not statistically significant**.
> > > - It's possible that their training setup (e.g., hyperparameter tuning) used in the original study [1] not be optimal. Since they don't give any details in their paper nor provide the code used to train NBFNet, it is hard to determine if this is the cause. It also limits our ability to replicate their results.
> > >
> > >
> > > Furthermore, in Table 5 we include the results across all models. On both datasets, the mean performance decreases (see table below). **This indicates, the the newer datasets are generally harder**.
> > >
> > > **Change in Performance Across All Methods**:
> > > | Dataset |  % Change |
> > > | ------- |  -------- |
> > > | WN18RR (E) | -40.6% |
> > > | FB15k-237 (E, R) | -9.5% |
> > >
> > >
> > > [1] Lee, Jaejun, et. al. "InGram: Inductive knowledge graph embedding via relation graphs." ICML, 2023.

---

> > > > ### Author Response · Authors · 2024-11-19
> > > > **Response to Reviewer wYLD (4/N)**
> > > >
> > > > > Why the performance difference of PPR and NBFNet on WN18RR dataset in both old and new versions are the same, but this is not the case for FB237?
> > > >
> > > > Our analysis suggests that this is because even on the transductive version of WN18RR, PPR can already perform well. Therefore, even when mitigating the shortcut, PPR can still achieve high performance and thus the relationship between it and NBFNet isn't modified much. On the other hand, this isn't true for FB15k-237.
> > > >
> > > > For clarity, below we show the % difference in performance of PPR vs. NBFNet on the old and new datasets. We further include the PPR performance on the old and new datasets.
> > > >
> > > > We can observe that on FB15k-237, the PPR performance decreases significantly to 10.8. **Since this is so low, it is much easier for NBFNet to outperform it, since there is a lot of room for improvement**. On the other hand, for WN18RR, the performance only drops to 45.1, which is still high.
> > > >
> > > > **PPR vs. NBFNet**:
> > > > | Metric | WN18RR (E) | FB15k-237 (E, R) |
> > > > | ------- | ------ | ---------- |
> > > > | Old % Change | +9.3% | 15.9% |
> > > > | New % Change | +16.4% | 149% |
> > > > | Old PPR Hits@10 | 66 | 21.4 |
> > > > | New PPR Hits@10 | 45.1 | 10.8 |
> > > >
> > > > Lastly, we'd like to note that for WN18RR, the new PPR performance is aligned with the transductive performance of 46.2. This is likely due to certain patterns specific to the original dataset that cause PPR to perform well.

---

> ### Author Response · Authors · 2024-11-25
> **Reminder about Rebuttal**
>
> Dear Reviewer wYLD,
>
> Thank you for taking the time to review our paper!
>
> As the discussion deadline is rapidly approaching, we are eager to hear your response on our rebuttal as soon as possible. We hope that all aspects of our response addresses your concerns and await additional questions.
>
> Regards,
>
> Authors

---

> > ### Comment · Reviewer_wYLD · 2024-11-25
> >
> > Thank the authors for their clarification. I have updated my score in light of their response to my review.

---

> > > ### Author Response · Authors · 2024-11-25
> > >
> > > Thank you very much!
> > >
> > > Please let us know if you have any other questions. We'll be happy to answer them.

---

### Official Review · Reviewer_TAPy · 2024-11-05

**Soundness:** 3
**Presentation:** 4
**Contribution:** 3
**Rating:** 8
**Confidence:** 4

**Summary:**

The main finding of the paper is that many existing inductive knowledge graph (KG) datasets are constructed in a way that creates significant differences in shortest path distances (SPD) between positive and negative samples. This discrepancy allows non-learning-based heuristics, like Personalized PageRank (PPR), which ignores relation types altogether and is unsuitable for knowledge graph completion (KGC), to achieve performance close to that of SOTA methods. This highlights that existing inductive KG datasets might be too simplistic, thereby failing to adequately test the full inductive capabilities of models.

The authors propose a novel dataset construction strategy using graph partitioning to ensure that the sampled subgraphs maintain the original KG structure while mitigating the introduction of SPD-based shortcuts. The proposed benchmark datasets are shown to be more challenging for inductive models, providing a better platform for evaluating their real capabilities.

**Strengths:**

1. **Addressing a Fundamental Problem in Existing Inductive Datasets**: The paper identifies that the shortest path distance (SPD) shortcut is prevalent among several existing inductive KG datasets. Furthermore, the authors demonstrate that all datasets generated by varying configurations of the existing data generation algorithm suffer from this problem. This indicates that the current inductive data generation strategy is fundamentally flawed, and addressing these flaws is crucial to genuinely evaluate a model's inductive capabilities.

2. **Well-Written and Clear Presentation**: The paper is well written and easy to follow. The problem statement, methodology, and experimental findings are presented in a clear and structured manner.

3. **Effective Remedial Strategy**: The authors propose an intuitive and effective strategy for sampling inductive KGs based on graph partitioning, ensuring that entities within a partition are densely connected while inter-partition connections are minimized. This maintains the original graph's SPD distribution, mitigating shortcuts and providing a fairer benchmark.

4. **Extensive Empirical Study**: The authors conducted an extensive empirical evaluation, providing a thorough analysis of both the existing and newly proposed datasets. The experiments cover a wide range of inductive KG completion methods, adding significant value to the paper.

**Weaknesses:**

**My main concern with this paper is that it misses key citations in Related Work**. Specifically, Gao et al. [1] was a concurrent work to Galkin et al. [2] that first provided a theoretical understanding of what is necessary for solving the (E, R) inductive KGC task. Gao et al. also introduced two new (E, R)-capable methods, ISDEA+ and DEq-InGram, with the latter being an improved version of the original InGram [3]. Additionally, two new (E, R) datasets, PediaTypes and WikiTopics, were proposed for empirical evaluation. Including discussions of all (E, R) capable methods, i.e. InGram (DEq-InGram), ULTRA, ISDEA+, in Section 2 (around lines 134-139) as well as including these datasets and methods in the empirical results would provide a more complete overview of recent advances in inductive KGC research and support a more comprehensive empirical comparison.

**Questions:**

1. **Analysis of Other (E, R) Inductive Datasets**: As discussed in W1, PediaTypes and WikiTopics are two fully inductive cross-domain KG datasets proposed by [1] that were not studied in this paper. Could you consider applying the same analysis to these datasets to determine whether they also exhibit divergent SPD differences between positive and negative samples? Additionally, could you evaluate the performance of PPR on these datasets? Including results for these datasets would significantly enhance the paper's comprehensiveness, particularly regarding its coverage of (E, R) inductive datasets.

2. **Evaluation of Other (E, R) Inductive Methods on New Inductive Dataset**: As discussed in W1, Gao et al. [1] proposed the (E, R) inductive methods ISDEA+ and DEq-InGram. Including these methods in the baseline experiments for Section 5 would provide a more comprehensive comparison with the other methods tested in the paper. Of particular interest is that the paper notes that "InGram struggles in the (E, R) setting" on the newly created dataset based on graph partitioning; therefore, it would be valuable to determine whether DEq-InGram could overcome these struggles or if it still faces similar challenges.

3. **Question about the Inference Setting on the (E, R) Inductive Experiments**: It seems that the results of ULTRA [2] on the new inductive datasets are placed in Figure 6, separate from the main (E, R) results shown in Table 4. Could you clarify why these results are presented separately? Is the distinction due to different inference settings (e.g., zero-shot inference for ULTRA in Figure 6 versus few-shot fine-tuning or test-time adaptation for other methods in Table 4)? If this is the case, explicitly mentioning this distinction in the experimental setup section would help reduce confusion and improve clarity.

**References**

[1] Gao, Jianfei, et al. "Double equivariance for inductive link prediction for both new nodes and new relation types." (2023)

[2] Galkin, Mikhail, et al. "Towards foundation models for knowledge graph reasoning." (2023)

[3] Lee, Jaejun, Chanyoung Chung, and Joyce Jiyoung Whang. "InGram: Inductive knowledge graph embedding via relation graphs." (2023)

---

> ### Author Response · Authors · 2024-11-19
> **Response to Reviewer (1/N)**
>
> Thank you for taking the time to review our paper. We reply to your questions below.
>
>
> > Evaluation of Other (E, R) Inductive Methods (ISDEA+ and DEq-InGram)
>
> We appreciate the reviewer bringing these methods to our attention.
>
> We used the official implementation of both methods. The results of each are over 5 random seeds. We'd like to clarify that in their paper, they only evaluate against 50 randomly selected negative entities. However, this diverges from the setting used in our paper and others (see ULTRA, InGram, RED-GNN) where we evaluate against all possible entities. To properly compare against the result in our paper, **we evaluate against all negative samples**.
>
> **(1) DEq-InGram**: We provide the results of DEq-InGram in the tables below for both (E) and (E, R) datasets broken down by inference graph. Furthermore, we include the rank of the method in parentheses, such that 1 means it is the best method. The full results are included in the revised paper in Tables 3 and 4.
>
> We observe that **(1)** It performs strongly in the (E, R) task **(2)** It improves upon standard InGram on each dataset. These results suggest that DEq-InGram can greatly improve the performance of InGram and achieve competitive performance on (E, R) datasets.
>
> | Datset | Inference Graph | Hits@10 | Performance Rank |
> | ------ | --------------  | ------- | ---------------  |
> | CoDEx-M (E) | 1 | 23.8 ± 1.6 | 3 |
> | WN18RR (E) | 1 | 62.5 ± 0.8 | 4 |
> | WN18RR (E) | 2 | 19.1 ± 3.1 | 4 |
> | HetioNet (E) | 1 | 26.5 ± 4.1 | 3 |
> | HetioNet (E) | 2 | 28.8 ± 3.5 | 3 |
> |          	|   |        	|   |
> | FB15k-237  (E, R) | 1 | 35.4 ± 2.5 | 1
> | FB15k-237  (E, R) | 2 | 27.1 ± 3.5 | 2 |
> | CoDEx (E, R) | 1 | 35.2 ± 14.4  | 2 |
> | CoDEx (E, R) | 2 | 24.7 ± 0.9 | 2 |
>
>
> **(2) ISDEA+**: Curiously, when running ISDEA+ **we found that when evaluating against all possible entities, we struggle to achieve performance marginally better than 0**. This applies both to our datasets and those used in the original ISDEA+ paper. We provide more analysis of this issue below.
>
> We first illustrate this problem on both the WN18RR (E) and FB15k-237 (E, R) datasets. Like detailed in their paper, we tune the learning rate from {1e-2, 1e-3, 1e-4}. They further mention tuning the weight decay from {5e-4, 0}, however we didn't find this in their code and thus omitted it. We further tuned the number of negatives per positive samples **in training** from {2, 16, 32}. In their paper they only train for 10 epochs, however we increase this to 200 epochs as the performance after 10 was nearly always 0.
>
> The results are shown below for Hits@10 when evaluating using all negatives and 50 random. We find that while the performance is very low when using all negatives, it is quite high when using 50.
>
> To ascertain if this is just a problem with our datasets, we further check the performance on the WikiTopics and PediaTypes datasets. We choose EN2FR-15K-V2 and wikidata_artv2 as representative datasets. For both datasets, we use the exact training settings provided by the authors (note that wikidata_artv2 is trained under the meta learning setting). **However, we observe the same issue as before, where the performance against all negative entities is extremely low**.
>
> Due to these issues, **we believe that there is a bug in their code, causing the performance against all negative entities to be artificially low**. Note that we are using their code, and have not modified the code for evaluation at all. **We have reached out to the authors with our findings and are waiting for a response**. We will update you if we are able to fix this issue.
>
> **ISDEA+ Performance on New Datasets:**
> | Dataset | Inference Graph | Hits@10 (All) | Hits@10 (50) |
> | ------ | --------------- | ------------- | ------------ |
> | WN18RR (E) | 1 | 1.7 | 90.3 |
> | WN18RR (E) | 2 | 1.3 | 72.1 |
> | FB15k-237  (E, R) | 1 | 0.6 | 74.3 |
> | FB15k-237  (E, R) | 2 | 2.5 | 61.7 |
>
> **ISDEA+ Performance on WikiTopics & PediaTypes:**
> | Dataset | Hits@10 (All) | Hits@10 (50) |
> | ------  | ------------- | ------------ |
> | EN2FR-15K-V2 | 0.5	| 96.3 |
> | wikidata_artv2 | 0.4	| 90.6 |
>
> > Including discussions of all (E, R) capable methods, i.e. InGram (DEq-InGram), ULTRA, ISDEA+, in Section 2 (around lines 134-139)
>
> We included a discussion of ISDEA+ and DEq-InGram in our revised paper. See lines 136-140 in red. For convenience, we also include the added component below:
>
> > *Gao et al. (2023) introduces the concept of “double permutation- equivariant representations” as a way to model KGs that are equivariant to permutations of both the entities and relations. They theoretically show that capturing this property is essential for proper generalization across KGs. To this point these introduce a new method ISDEA/ISDEA+ that can satisfy this property. They further introduce a variant of InGRAM Lee et al. (2023), DEq-InGram, that endows it with the ability to compute double equivariant representations.*

---

> ### Author Response · Authors · 2024-11-19
> **Response to Reviewer (2/N)**
>
> > Analysis of Other (E, R) Inductive Datasets two new (E, R) datasets (PediaTypes and WikiTopics)
>
>
> We include the analysis for both sets of datasets. As before, the performance is Hits@10 and for predicting the correct node (h, r, ?). For each dataset we measure:
> - **Performance of ISDEA+ on 50 negatives**. The performance is taken from the paper. This is Table 1b for PediaTypes and Table 7b for WikiTopics (we use the best here). For WikiTopics, the ISDEA+ performance isn't reported for three datasets.
> - **Performance of PPR on 50 negatives**
> - **% Difference in performance (50 negatives)**. Note that a negative value means PPR performs better
> - **Performance of PPR when evaluating on all negatives** . This allows us to compare our paper's results.
> - The **$\Delta$ SPD of each dataset.**
>
> The results are in the two tables below, where we group the PediaTypes and WikiTopics datasets separately. We also include the mean of each column as the final row.
>
> We make several observations:
> 1. **The difference in performance (50 negatives) between ISDEA+ and PPR is very small** across all datasets. Across the 8 PediaTypes datasets, ISDEA+ only performs 0.4% better on average. For WikiTopics, this number is 14.2%. This is in fact lower than what we found in our paper for an inductive dataset, where neural methods performed 29% better than PPR.
> 2. **The $\Delta$ SPD of these datasets is high**. It is on average 2.2 and 2.7 for PediaTypes and WikiTopics, respectively. This is higher than that of the transductive datasets used in our paper (1.2 on average) and more in line with inductive datasets (4.7 across all and 2.1 when WN is excluded as it features very high values).
> 3. **The PPR performance on all negatives is very high**. On average, it is 32.5% and 38.3% on PediaTypes and WikiTopics, respectively. This is similar to the inductive datasets used in our study, where the average PPR performance is 35.6%.
>
> **These results show that these datasets suffer from the same shortcut found in our paper**. This is as expected, as the PediaTypes datasets are constructed using the traditional method subgraph-based described in our paper. Also, WikiTopics uses the Forest Fire [1] sampling method, which generally favors locality, and thus will have the same downsides as the subgraph-based method.
>
> We have revised our paper to include these results. See Appendix I.
>
>
> **PediaTypes Results:**
> | Dataset	| ISDEA+ Hits@10 (Neg=50)	| PPR Hits@10 (Neg=50)	| Hits@10 % Diff (Neg=50)	| PPR Hits@10 (Neg=All) | $\Delta$ SPD |
> | -------   | --------------------  | ------------  | ------------  |  ------------  | ------------- |
> | PediaTypes EN-FR | 95.4%	| 96.0%	| -0.6%	| 44.1% | 2.7
> | PediaTypes FR-EN | 90.6%	| 96.2%	| -5.8%	| 48.1% | 2.3
> | PediaTypes EN-DE | 97.7%	| 93.9%	| 4.0%	| 34.7% | 2.1
> | PediaTypes DE-EN | 95.0%	| 90.6%	| 4.9%	| 20.8% | 1.6
> | PediaTypes DB-WD | 86.6%	| 76.4%	| 13.4%	| 23.5% | 0.6
> | PediaTypes WD-DB | 91.5%	| 92.0%	| -0.5%	| 42.9% | 2.5
> | PediaTypes DB-YG | 71.5%	| 76.9%	| -7.0%	| 15.2% | 2.7
> | PediaTypes YG-DB | 80.5%	| 85.1%	| -5.4%	| 30.7% | 3.5
> | **MEAN**         	|   88.6%	| **88.4%**	| **0.4%**	| **32.5%** | **2.2**
>
>
> **WikiTopics Results:**
> | Dataset	| ISDEA+ Hits@10 (Neg=50)	| PPR Hits@10 (Neg=50)	| Hits@10 % Diff (Neg=50)	| PPR Hits@10 (Neg=All) | $\Delta$ SPD |
> | -------   | --------------------  | ------------  | ------------  |  ------------  | ------------  |
> | WikiTopics Art	| 92.3%	| 79.3%	| 16.4%	| 14.7% | 1.1
> | WikiTopics Award	| 93.6%	| 95.8%	| -2.3%	| 24.2% | 3.4
> | WikiTopics Edu	| 97.0%	| 72.4%	| 34.0%	| 20.8% | 2.7
> | WikiTopics Health	| 98.9%	| 91.5%	| 8.1%	| 59.6% | 3.3
> | WikiTopics Infra	| 97.8%	| 97.2%	| 0.6%	| 61.4% | 5.4
> | WikiTopics Sci	| 96.7%	| 84.3%	| 14.7%	| 25.5% | 4.4
> | WikiTopics Sport	| 81.2%	| 82.9%	| -2.1%	| 14.4% | 0.3
> | WikiTopics Tax	| 93.7%	| 64.1%	| 46.1%	| 29.0% | 0.8
> | WikiTopics Loc	| N/A	| 96.5%	| N/A | 74.1% | 1.4
> | WikiTopics Org	| N/A	| 94.0%	| N/A | 8.9% | 2.8
> | WikiTopics People	| N/A	| 94.9%	| N/A |	64.6% | 2.7
> | **MEAN**         	|   94.1%	| **87.4%**	| **14.2%**	| **38.3%** | **2.7**
>
>
> [1] Leskovec, Jure, Jon Kleinberg, and Christos Faloutsos. "Graph evolution: Densification and shrinking diameters." TKDD, 2007.
>
> > It seems that the results of ULTRA [2] on the new inductive datasets are placed in Figure 6, separate from the main (E, R) results shown in Table 4. Could you clarify why these results are presented separately?
>
> We apologize for the confusion.
>
> As the reviewer noted, ULTRA is presented separately due to the fact that ULTRA is zero-shot while the other methods in Table 4 are not. This allows us to properly delineate between the type of methods. To make this clearer, we revised the paper to include this note, specifically we included in red on lines 520-522.
>
> > *Since the setting of ULTRA differs from that of the other methods (i.e., 0-shot), we display it separately from the other methods in Tables 3 and 4.*

---

> > ### Comment · Reviewer_TAPy · 2024-11-23
> >
> > Thanks for the authors' thorough rebuttal and great efforts at the additional experiments and analysis. I found it particularly interesting that the authors were able to find consistent shortcut problems on the new (E, R) datasets I suggested. The authors have fully addressed all my concerns.
> >
> > I believe the updated paper in its current form is well-rounded and has the potential to make a significant impact on the field of KGC. The work tackles a fundamental and prevalent issue with the way we evaluate the current methods. The proposed graph partitioning strategy, while seemingly simple, is both effective and easy to implement, and fixes an important gap in the literature.
> >
> > Based on these improvements, I am happy to raise my score and recommend this work for acceptance.

---

> ### Author Response · Authors · 2024-11-23
>
> Thank you very much!
>
> Please let us know if you have any other questions. We'll be happy to answer them.

---

### Meta-Review · Area_Chair_gtjt · 2024-12-17

**Metareview:**

This work finds that in terms of Hits@10, simply using Personalized PageRank (PPR) can achieve relatively high performance on existing inductive knowledge graph completion (KGC) datasets. To resolve this issue, the authors propose a new strategy for constructing inductive KGC datasets, which utilizes graph partitioning algorithms such as Spectral Clustering or Louvain method. The authors provide new experimental results on their proposed datasets for state-of-the-art inductive methods such as InGram, RED-GNN, and NBFNet.

Unfortunately, the authors violated the anonymity policy. In install.md, there is a code written: "git clone git@github.com:**HarryShomer**/Better-Inductive-KGC.git" (https://anonymous.4open.science/r/KG-PPR-ICLR-D0B7/install.md), which reveals one of the authors' GitHub repository: https://github.com/HarryShomer

Since this link was provided within the anonymized link, it wasn't easy to find this during the desk-reject period. When I tried to reproduce the results myself, I found the install.md file contains this identification information. Therefore, it is likely that the reviewers also know the author's information, which can have affected the review process.

I have also thoroughly read the manuscript and found some critical points described below.
1. Analysis only on Hits@10\
In KGC literature, several standard metrics exist: Mean Reciprocal Rank (MRR), Hits@10, (Hits@5, Hits@3,) and Hits@1. Each of these metrics shows a different perspective of a method. At least, MRR, Hits@10, and Hits@1 can be considered essential metrics to consider. However, this manuscript only mentions Hits@10 and ignores the other metrics. The authors should show their results using diverse evaluation metrics.

2. Insufficient Hyperparameter Tuning for RED-GNN, NodePiece, and InGram\
In the RED-GNN paper, it is mentioned that RED-GNN was tuned on 2880 hyperparameter sets (Section 5). Also, in the InGram paper, the authors mentioned 6912 hyperparameter sets (Appendix C). However, the authors of this manuscript tuned RED-GNN, NodePiece*, and InGram only using 4 hyperparameter sets. Therefore, it is likely that the unsatisfactory results of RED-GNN, NodePiece, and InGram reported in this manuscript can be due to insufficient hyperparameter tuning for these methods. Not using a reasonable number of hyperparameters (and thus potentially making the methods under-tuned) is a critical point and introduces risks of making wrong conclusions. (*NodePiece does not explicitly mention the range of hyperparameters in the original paper, but it is likely that more than 4 hyperparameter sets are considered in their experiments.)

3. Different Experimental Settings for NodePiece and InGram\
This manuscript's experimental setting assumes that the validation set is from the train graph. However, NodePiece and InGram assume a different experimental setting: the validation set is from the inference graph to reduce the gap between the validation and test performance (note that the validation and test sets are disjoint). Even though these different experimental settings can critically affect the models' performance, the authors have modified them without mentioning them in the manuscript. Due to these modifications, the performance of NodePiece and InGram reported in the manuscript is unreliable. The authors should correct the experimental settings for NodePiece and InGram to be identical to their original papers and re-do the experiments.

**ULTRA is in ICLR 2024, not 2023. Please check the reference.

**Additional Comments On Reviewer Discussion:**

As mentioned in the metareview, the author's information was revealed through the link provided by the authors, which violates the anonymity policy, and this leaking of information has hindered fair evaluation. Additionally, the manuscript has critical flaws and drawbacks, which are also elaborated in the metareview.

---

### Decision · Program_Chairs · 2025-01-22

Reject